# Natural forgetting reversibly modulates engram expression

**James D O'Leary[1,2], Rasmus Bruckner[2,3,4], Livia Autore[1,2], Tomás J Ryan[1,2,5,6]***

[1]School of Biochemistry and Immunology, Trinity College Dublin, Dublin, Ireland; [2]Trinity College Institute of Neuroscience, Trinity College Dublin, Dublin, Ireland; [3]Department of Education and Psychology, Freie Universität Berlin, Berlin, Germany; [4]Max Planck Research Group NeuroCode, Max Planck Institute for Human Development, Berlin, Germany; [5]Florey Institute of Neuroscience and Mental Health, Melbourne Brain Centre, University of Melbourne, Melbourne, Australia; [6]Child & Brain Development Program, Canadian Institute for Advanced Research (CIFAR), Toronto, Canada

## eLife assessment

This is an **important** paper on the role of engrams and relevant conditions that influence memory and forgetting. The variety of methods used, namely, behavioural, labeling, interrogation, immuno-histochemistry, microscopy, pharmacology, computational, are exemplary and provide **convincing** evidence for the role of engrams in the dentate gyrus in memory retrieval and forgetting. This examination will be of interest broadly across behavioural and neural science communities.

***For correspondence:**
tomas.ryan@tcd.ie

**Abstract** Memories are stored as ensembles of engram neurons and their successful recall involves the reactivation of these cellular networks. However, significant gaps remain in connecting these cell ensembles with the process of forgetting. Here, we utilized a mouse model of object memory and investigated the conditions in which a memory could be preserved, retrieved, or forgotten. Direct modulation of engram activity via optogenetic stimulation or inhibition either facilitated or prevented the recall of an object memory. In addition, through behavioral and pharmacological interventions, we successfully prevented or accelerated forgetting of an object memory. Finally, we showed that these results can be explained by a computational model in which engrams that are subjectively less relevant for adaptive behavior are more likely to be forgotten. Together, these findings suggest that forgetting may be an adaptive form of engram plasticity which allows engrams to switch from an accessible state to an inaccessible state.

## Introduction

For an organism to adapt to its environment there must be processes by which learned information is updated or replaced with new more relevant information (*Pignatelli et al., 2019*; *Cowan et al., 2021*). For example, learning that a specific food source is no longer present in a previous location, or that a given action stops yielding the same outcome (*Pignatelli et al., 2019*; *Hughes et al., 1992*; *Eliassen et al., 2007*). Without a process to update or forget irrelevant information, adaptability would be impaired with frequent instances of correct but outdated memory recall, and a subsequently biased behavioral response (*Cowan et al., 2021*; *Eliassen et al., 2007*; *Marvin and Shohamy, 2016*; *Den Ouden et al., 2009*; *Miller, 2021*). Forgetting, i.e., the loss of learned behavioral responses, may therefore function adaptively to improve learning and promote survival (*Ryan and Frankland, 2022*).

Memories are stored as ensembles of engram cells that undergo specific forms of plasticity during learning, and their successful recall involves the reactivation of these cellular networks (*Josselyn and Tonegawa, 2020*; *Ortega-de San Luis and Ryan, 2022*; *Tonegawa et al., 2015*; *Josselyn et al., 2015*; *Denny et al., 2017*). However, despite advancements in understanding the biology of engram cells, more work is still needed to fully elucidate how these cell ensembles contribute to the process of forgetting (*Ryan and Frankland, 2022*; *Denny et al., 2017*; *Richards and Frankland, 2017*). Forgetting is often considered a deficit of memory function. The standard view in the field of memory is that this kind of active forgetting is due to the loss or dissipation of the engram/trace (*Ryan and Frankland, 2022*). We argue that a new conceptualization is necessitated by recent behavioral and physiological findings that emphasize retrieval deficits as a key characteristic of memory impairment, supporting the idea that memory recall or accessibility may be driven by learning feedback from the environment (*Miller, 2021*; *Ryan and Frankland, 2022*; *Richards and Frankland, 2017*; *Perusini et al., 2017*; *Guskjolen et al., 2018*; *Roy et al., 2016*; *Autore et al., 2023*). Little is understood about the role experience plays in altering forgetting rates. Recently it has been suggested that different forms of forgetting may exist along a gradient of engram activity or expression (*Ryan and Frankland, 2022*). Indeed, numerous studies have suggested that memory recall (*Perusini et al., 2017*; *Roy et al., 2016*; *Frankland et al., 2019*; *Josselyn and Frankland, 2018*; *Lei et al., 2022*; *Lv et al., 2019*), and most recently forgetting, require engram activity (*Autore et al., 2023*; *Khalaf et al., 2018*). Long-term memories of salient experiences can last a lifetime and must involve significant and specific changes to brain structure, while the successful recall of learned information requires the reactivation of these cell ensembles (*Tonegawa et al., 2015*). In contrast, forgetting may occur when engrams are not or cannot be reactivated (*Ryan and Frankland, 2022*; *Autore et al., 2023*; *Poo et al., 2016*). In severe amnesic states, this may be due to the destruction of the engram itself (*Ryan and Frankland, 2022*; *Washington et al., 2016*; *Chapman et al., 2021*; *Sloley et al., 2021*; *McTighe et al., 2010*). However, in other pathological and non-pathological states, forgetting may be due to reduced accessibility of engrams that otherwise endure (*McTighe et al., 2010*). Altered engram accessibility may be caused by structural plasticity in the engram or indeed by competing engrams of similar or recent experiences (*Ryan and Frankland, 2022*; *Autore et al., 2023*; *Davis and Zhong, 2017*; *Wang et al., 2020*; *Rao-Ruiz et al., 2019*; *Rashid et al., 2016*; *Poll et al., 2020*).

We hypothesize that natural forgetting represents a reversible suppression of engram ensembles due to experience and perceptual feedback, prompting cellular plasticity processes that modulate memory access adaptively. To investigate this idea, we utilized a forgetting paradigm based on an object recognition task and interrogated the conditions in which a memory could be preserved, retrieved, or forgotten to elucidate how experience and learning govern engram reactivation and memory recall. Based on our experimental results, we used the Rescorla-Wagner model to explain our findings across studies in a formal framework. This model describes forgetting as a learning process through which engram accessibility is dynamically updated in response to perceptual feedback, and engrams that are subjectively less relevant for adaptive behavior are more likely to be 'forgotten'.

## Results

### Natural forgetting of an object memory

Natural forgetting was assessed using an object recognition task. This task utilizes the natural tendency of mice to explore novel stimuli to assess recognition memory over time (*Figure 1a*, *Bevins and Besheer, 2006*; *Leger et al., 2013*). During object memory recall, mice tested 24 hr after training spent significantly more time exploring the novel object compared to the familiar object, $t(9) = 5.12$, $p<0.001$, 95% CI [11.18–28.68]. Mice that were tested 2 weeks after training spent a similar amount of time exploring both novel and familiar objects, $t(11) = 0.93$, $p=0.36$, 95% CI [–3.88 to 9.64] (*Figure 1b*). In order to compare performance in object memory, a discrimination index was calculated. The discrimination index was calculated as the time spent exploring the novel object minus the time spent exploring the familiar object divided by the total time spent exploring both objects (novel–familiar/novel+familiar). There was a significant difference between the discrimination index of the 24 hr and 2 week group, $t(20) = 4.30$, $p<0.001$, 95% CI [–0.59 to –0.20] (*Figure 1c*). This finding was further supported by a minute-by-minute analysis of object exploration (*Figure 1—figure supplement 1*). Together, these data indicate that mice successfully recall an object memory 24 hr

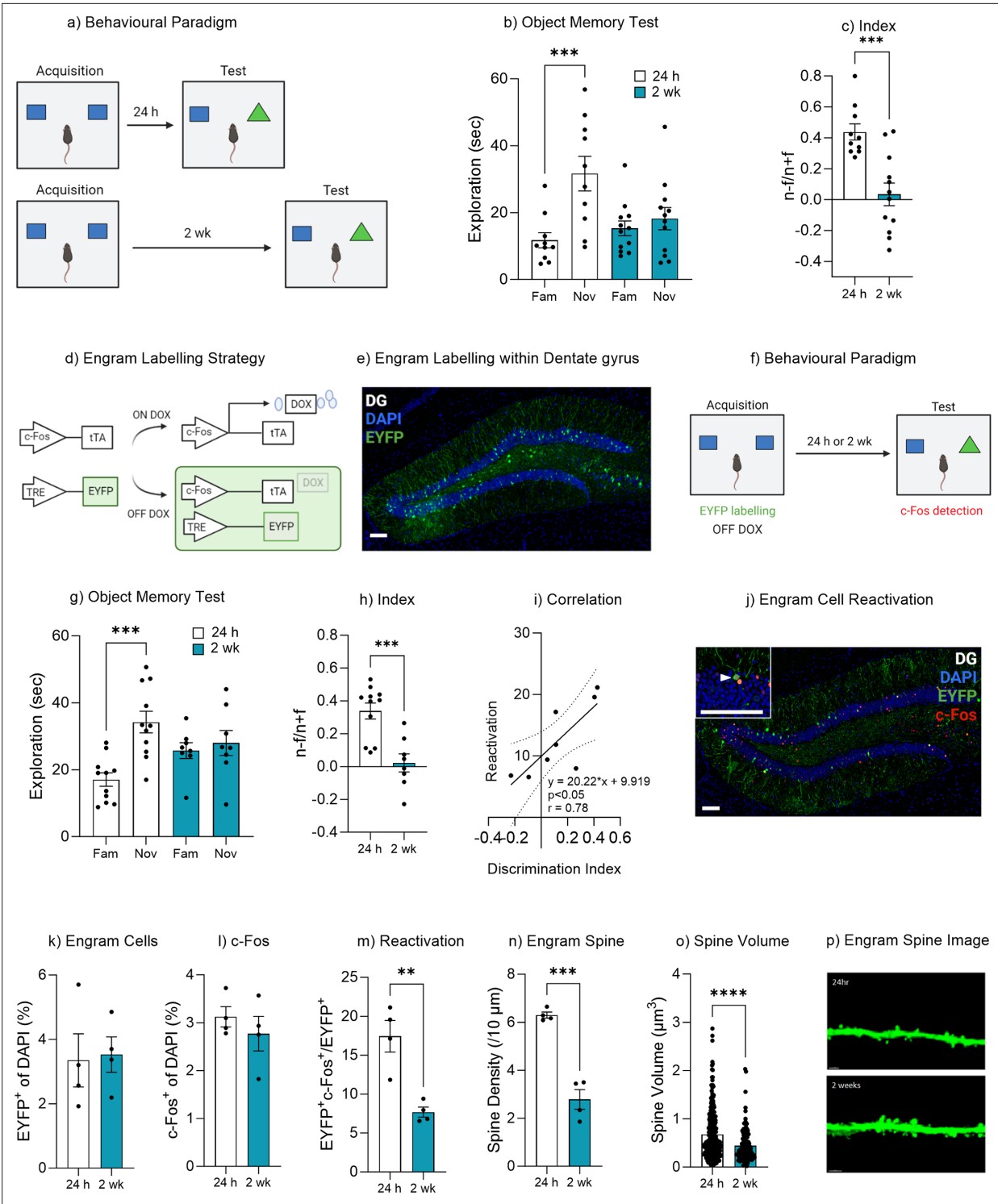

**Figure 1.** Natural forgetting of an object memory. (**a**) Object recognition paradigm. (**b**) Object memory test for recall at 24 hr or 2 weeks. (**c**) Discrimination index. (**d**) Schematic of Fos-tTA engram labeling system. (**e**) Representative image of engram labeling within the dentate gyrus. (**f**) Engram labeling of object memory and c-Fos detection following recall. (**g**) Object memory test for recall at 24 hr or 2 weeks. (**h**) Discrimination index. (**i**) Correlation between discrimination index and engram reactivation. (**j**) Representative image eYFP+ cells, c-Fos+ cells and merged eYFP+ and c-Fos+ for both 24 hr and 2 week test. (**k**) Engram cells. (**l**) c-Fos+ cells. (**m**) Engram reactivation. (**n**) Engram spine density average per mouse. (**o**) Engram spine volume average per dendrite. (**p**) Representative image of engram dendrite for morphological analysis. Bar graphs indicate average values in $n$=4–11 per group (**p<0.01, ***p<0.001). Data graphed as means ± SEM. Scale bar 100 μm. Panel a was created with BioRender.com. Panel f was created with BioRender.com.

*Figure 1 continued on next page*

*Figure 1 continued*

The online version of this article includes the following figure supplement(s) for figure 1:

**Figure supplement 1.** Object memory time bin analysis.

after acquisition, while at 2 weeks there was no preference for the novel object suggesting the original familiar object had been forgotten.

## Engram reactivation is associated with object memory recall

In order to assess engram reactivation on object memory retrievability an AAV$_9$-TRE-ChR2-eYFP virus was injected into the dentate gyrus of Fos-tTA mice (*Reijmers et al., 2007*; *Ramirez et al., 2013*). The immediate early gene *Fos* becomes upregulated following neural activity and as such the Fos-tTA transgene selectively expresses tTA in active cells (*Ramirez et al., 2013*; *Ramirez et al., 2015*). The activity-dependent tTA induces expression of ChR2-eYFP following a learning event (*Figure 1d and e*). To restrict activity-dependent labeling to a single learning experience, mice were maintained on a diet containing doxycycline which prevents binding of tTA to the TRE elements of the TetO promoter, thus controlling the activity-dependent labeling to a specific time window of the object acquisition training (*Figure 1d–f*). Prior to training (36 hr) doxycycline was removed from the animal's diet, thereby allowing for activity-dependent labeling during the subsequent acquisition training (*Figure 1f*). During the object recall test, Fos-tTA mice tested 24 hr after acquisition training spent significantly more time exploring the novel object compared to the familiar object, t(10) = 6.00, p<0.001, 95% CI [10.80–23.54]. Mice that were tested 2 weeks after training spent a similar amount of time exploring both novel and familiar objects, t(7) = 0.67, p=0.51, 95% CI [–5.69 to 10.27] (*Figure 1g*). In order to compare performance in object memory, a discrimination index was calculated, as described earlier. There was a significant difference between the discrimination index of the 24 hr and 2 week group, t(17) = 4.30, p<0.001, 95% CI [–0.47 to –0.16] (*Figure 1h*). When comparing the engram ensemble between the two cohorts, we observed no significant difference in the number of engram cells between the 24 hr and 2 week tested groups (eYFP$^+$), t(6) = 0.18, p=0.86, 95% CI [–2.24 to 2.60] (*Figure 1j and k*). Similarly, there was no significant difference in c-Fos activity within the dentate gyrus between the 24 hr tested or 2 week tested mice, t(6) = 0.84, p=0.42, 95% CI [–1.37 to 0.66] (*Figure 1l*). However, when comparing engram reactivation, i.e., the number of engram cells (eYFP$^+$) that also expresses c-Fos during the recall test, which indicates these cells were active during encoding and then became active again during the memory recall (*Figure 1j*), there was a significant decrease in engram reactivation in mice that were tested 2 weeks after training and that had forgotten the object memory, t(6) = 4.56, p=0.003, 95% CI [–14.96 to –4.52] (*Figure 1m*). Furthermore, the level of engram reactivation correlated with the performance in object discrimination, r(1, 6) = 0.61, p=0.02 (*Figure 1i*). Together these data suggest that natural forgetting may be driven by a reduction in engram ensemble activity.

## Engram spine density decreased following natural forgetting

In order to gain further insight into the changes that occur within the memory engram following natural forgetting we performed morphological analysis of engram dendritic spines following successful memory recall at 24 hr or forgetting at 2 weeks. Following natural forgetting mice displayed a significant reduction in spine density, t(6) = 8.23, p<0.001, 95% CI [–4.55 to –2.46] (*Figure 1n*). Moreover, there was also a significant decrease in spine volume, t(372) = 4.50, p<0.001, 95% CI [–0.33 to –0.13] (*Figure 1o and p*). Together, these findings suggest that in addition to the reduced engram activity, there was also a reduction in engram spine density.

## Dentate gyrus engrams are necessary and sufficient for recall of an object memory

To test the hypothesis that the retrieval of an object memory requires the activity of engram cells within the dentate gyrus, we labeled engrams with the virus AAV$_9$-TRE-ArchT-GFP and inhibited engram activity during memory retrieval 24 hr following acquisition training (*Figure 2a*). During the object recall test the No Light control mice spent significantly more time with the novel object compared to the familiar object, indicating successful memory retrieval, t(9) = 2.91, p=0.01, 95% CI [3.43–27.09]

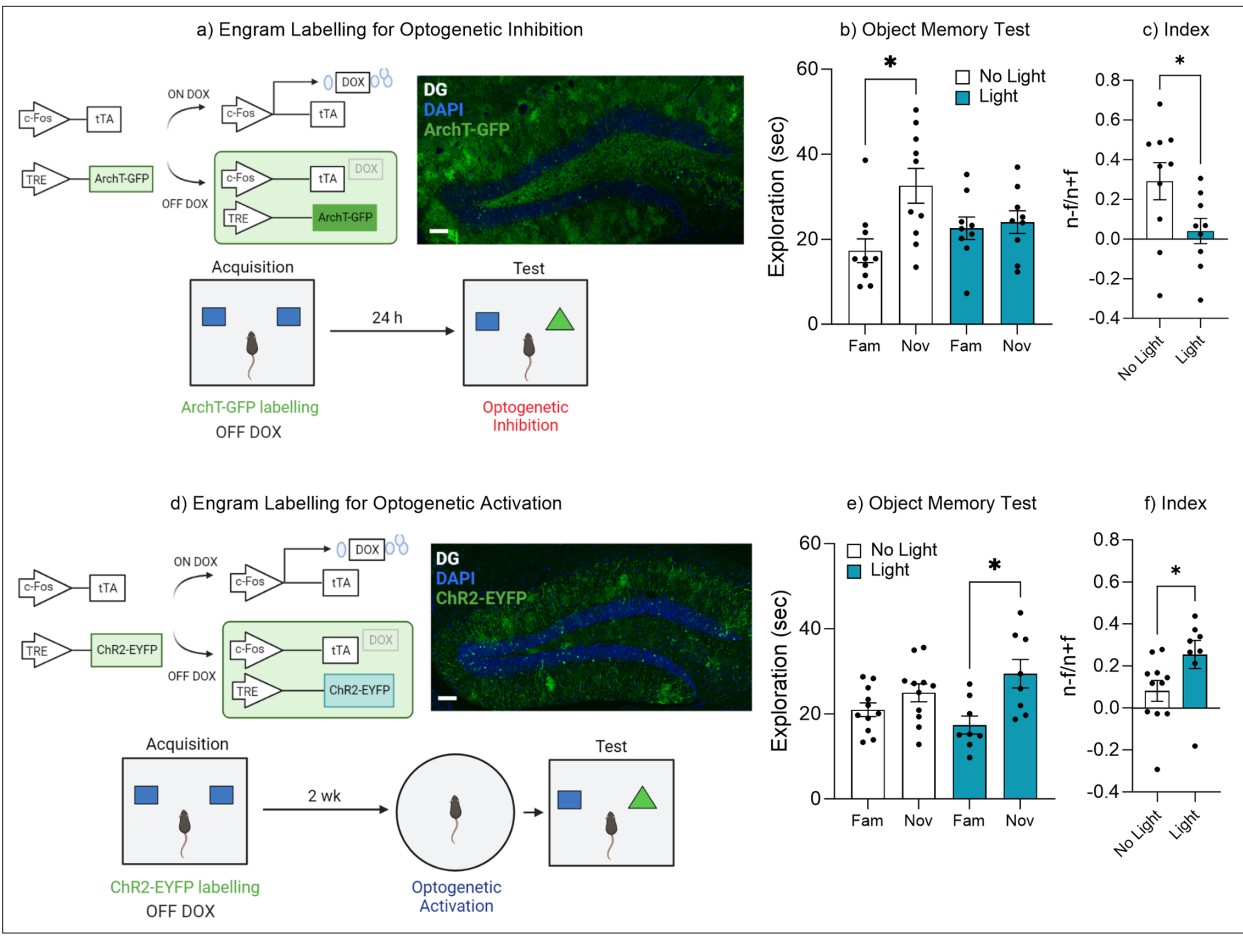

**Figure 2.** Dentate gyrus engrams are necessary and sufficient for recall of an object memory. (**a**) Engram labeling for optogenetic inhibition and behavioral timeline. (**b**) Object memory test. (**c**) Discrimination index. (**d**) Engram labeling for optogenetic activation and behavioral timeline. (**e**) Object memory test. (**f**) Discrimination index. Bar graphs indicate average values in *n*=9–12 per group (*p<0.05). Data graphed as means ± SEM. Scale bar 100 µm. Panel a was created with BioRender.com. Panel d was created with BioRender.com.

(*Figure 2b*). Whereas mice in which the engram was inhibited spent equal time exploring both the familiar and novel objects, indicating mice failed to distinguish the novel from the familiar object, t(8) = 0.58, p=0.57, 95% CI [–4.21 to 7.09] (*Figure 2b*). In order to compare performance in object memory, a discrimination index was calculated, as described earlier. There was a significant difference between the discrimination index of the No Light control and Light-induced inhibition groups, t(17) = 2.18, p=0.04, 95% CI [–0.49 to –0.008] (*Figure 2c*). Together, these data demonstrate that the engram cells within the dentate gyrus are required for successful retrieval of an object memory.

Since the data suggested that forgetting correlated with a reduction in engram activity (*Figure 1*) and that inhibition of the ensemble was sufficient to impair successful recall of an object memory (*Figure 2b and c*), we next asked if the retrieval of an object memory could be induced via activation of the engram after natural forgetting. We labeled engrams cells within the dentate gyrus with the virus AAV$_9$-TRE-ChR2-eYFP and optogenetically activated the ensemble after natural forgetting. Specifically, optogenetic stimulation was induced just prior to memory recall (3 min), but not during the recall test itself (*Figure 2d*). During the object recall test the No Light control mice displayed equal exploration of both the novel and familiar objects suggesting mice had forgotten the familiar object, t(10) = 1.81, p=0.10, 95% CI [–0.90 to 8.84] (*Figure 2e*). Whereas mice that underwent optogenetic activation spent more time exploring the novel object compared to the familiar object, suggesting the original object memory had been retrieved, t(7) = 3.41, p=0.01, 95% CI [3.71–20.43] (*Figure 2e*). In order to compare performance in object memory, a discrimination index was calculated. There was a significant difference between the discrimination index of the No Light control and Light-induced activation group, t(17) = 2.14, p=0.04, 95% CI [0.002–0.34] (*Figure 2f*). These data suggest that artificial

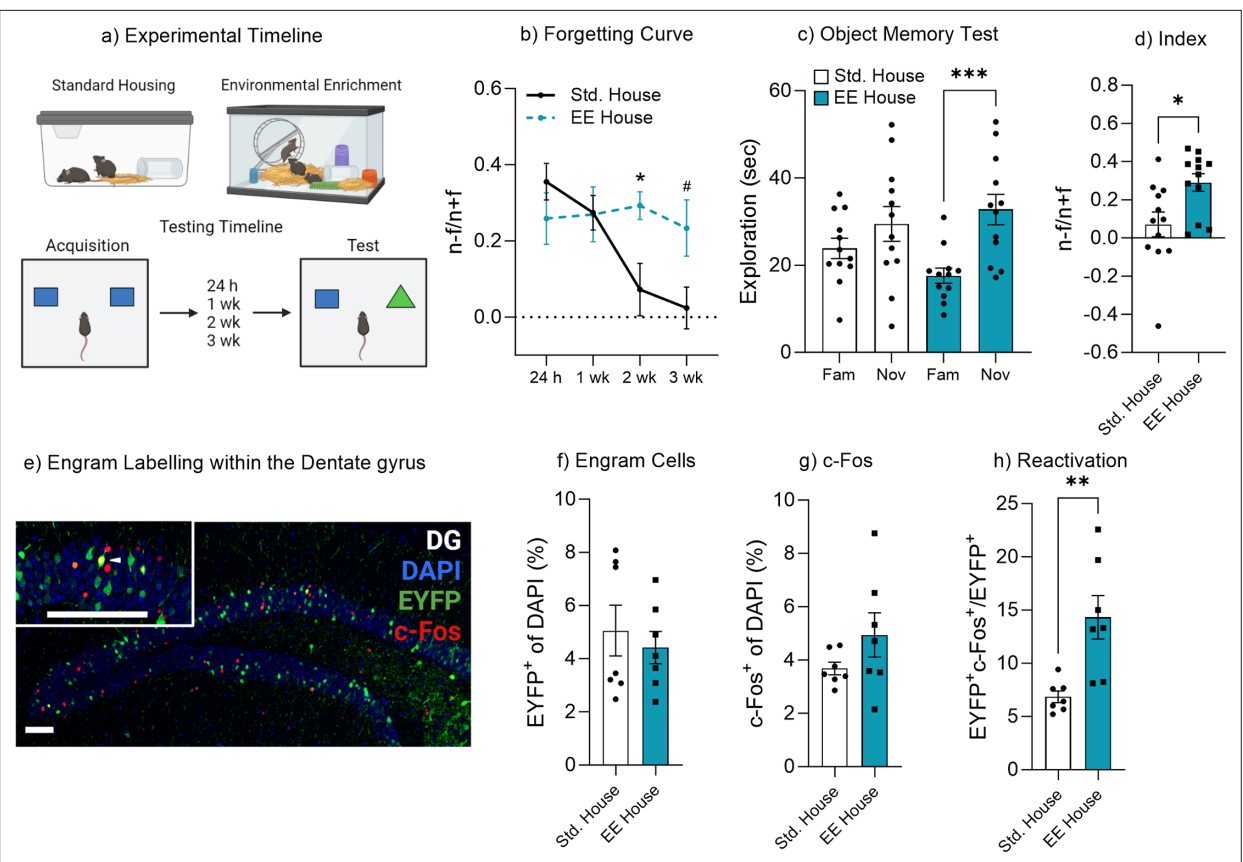

**Figure 3.** Environmental enrichment reduces the rate of forgetting by increasing engram activation. (**a**) Enrichment behavioral paradigm. (**b**) Natural forgetting curve of an object memory at 24 hr, 1 week, 2 week, and 3 week. (**c**) Object memory test. (**d**) Discrimination index. (**e**) Engram labeling within the dentate gyrus. (**f**) Engram cells. (**g**) c-Fos$^+$ cells. (**h**) Engram reactivation. Bar graphs indicate average values in *n*=4–12 per group (*p<0.05, ***p<0.001). Data graphed as means ± SEM. Scale bar 100 µm. Panel a was created with BioRender.com.

The online version of this article includes the following figure supplement(s) for figure 3:

**Figure supplement 1.** Object memory time bin analysis for environmental enrichment (EE).

**Figure supplement 2.** Environmental enrichment increased hippocampal neurogenesis.

activation of the original engram was sufficient to induce recall of the forgotten object memory, and that the forgotten engram was still intact, otherwise activation of those cells would no longer trigger memory recall. Together, the results of *Figure 2* suggest that engram activity can modulate memory retrieval, where activation of the engram ensemble is both necessary for successful memory retrieval and sufficient to induce recall despite natural forgetting.

## Environmental enrichment reduces the rate of forgetting by increasing engram activation

Environmental enrichment has been shown to enhance cognitive function and improve memory retention (*Bekinschtein et al., 2011*; *Gubert and Hannan, 2019*; *Clemenson et al., 2015*; *O'Leary et al., 2019a*; *O'Leary et al., 2019b*). A key component of enriched environments is the additional perceptual and tactile feedback provided by extra toys and objects within the home cage for the mice to interact with and explore. We therefore sought to investigate how enrichment might alter the rate of forgetting through changes in the engram given this increase in tactile and perceptual experience. First, we characterized the forgetting curve of mice housed under either standard or enriched conditions (*Figure 3a*). There was a significant effect of enriched housing on object memory performance F(3, 53)=3.26, p=0.028, 95% CI [–0.18 to 0.01] (*Figure 3b*). Bonferroni post hoc comparisons indicated there was no difference in object memory performance between the standard housed (Std) or environmental enrichment housed (EE) mice when tested 24 hr or 1 week after acquisition training,

indicating no change in memory recall (*Figure 3b*). However, there was a significant difference in discrimination index between standard housed and enriched mice when tested 2 hr after training (p=0.047), suggesting the enriched housed mice were still able to recall the familiar object, whereas the standard housed mice had forgotten. This finding was further supported by a minute-by-minute analysis of object exploration (*Figure 3—figure supplement 1*).

In order to assess the impact of environmental enrichment on engram activity we again injected an AAV$_9$-TRE-ChR2-eYFP virus into Fos-tTA mice to label engram cells within the dentate gyrus. Following surgery, mice were housed in either standard or enriched housing for 3 weeks before behavioral testing and remained in the housing condition for the duration of the experiment. Again, we observed that mice housed in enrichment maintained the object memory 2 weeks after training, as they spent more time exploring the novel object compared to the familiar object, t(11) = 5.05, p<0.001, 95% CI [8.59–21.83]. Mice housed in standard housing spent a similar amount of time exploring both the novel and familiar objects, t(11)=1.77, p=0.10, 95% CI [–1.34 to 12.57] (*Figure 3c*). In order to compare performance in object memory, a discrimination index was calculated. There was a significant difference between the discrimination index of standard housed (Std) and enriched (EE) mice, t(22) = 2.75, p=0.01, 95 % CI [0.054–0.38] (*Figure 3d*). When comparing the engram size between standard and enriched mice, we observed no significant difference in the number of engram cells following enrichment, t(12) = 0.56, p=0.58, 95% CI [–3.09 to 1.82] (*Figure 3e and f*). There was a marginal, but non-significant increase in the number of labeled cells within the standard housed mice in *Figure 3f*. The cell counting criteria was the same across experimental groups and conditions, where the entire dorsal and ventral blade of the dorsal DG was counted for each animal. Similarly, there was also no significant difference in c-Fos activity, t(12) = 1.45, p=0.17, 95% CI [–0.62 to 3.13] (*Figure 3g*). However, there was a significant increase in engram reactivation in mice that underwent enrichment, t(12) = 3.52, p=0.004, 95% CI [2.84–12.06] (*Figure 3h*). These data suggest that environmental enrichment reduces the rate of forgetting, by increasing or maintaining engram reactivation. To confirm the effectiveness of the enrichment paradigm to enhance neural plasticity, we measured hippocampal neurogenesis following environmental enrichment. Enrichment increased hippocampal neurogenesis, with both the number of immature neurons (doublecortin$^+$ cells) (*Figure 3—figure supplement 2*) and neuronal survival (BrdU-NeuN-positive cells) (*Figure 3—figure supplement 2*). The effects of enrichment on neural plasticity are well established and it likely contributes to the enhanced memory recall. In addition, the enriched environment, as it contains extra items such as objects and toys, may also help to reinforce the original object memory, resulting in the preservation of an object's memory through multiple engram retrievals, and updating plasticity. In this framework, the additional reminding experience of enrichment toys might help maintain or nudge mice to infer a higher engram relevancy that is more robust against forgetting, leading to higher engram expression.

## Exposure to the original stimuli facilitates memory recall

We next sought to investigate conditions in which more nuanced learning from experience may alter engram activity and subsequently switch a forgotten memory from an inaccessible state to an accessible state. Previous work from both human experimental psychology studies and rodent behavioral paradigms have shown that a brief exposure to reminder cues can aid memory recall (*Wahlheim et al., 2019*; *Tambini et al., 2017*; *Finkelstein et al., 2022*; *Bouton, 1993*). Here, we modified our object-based paradigm to include a brief exposure to the original encoding environment and objects (*Figure 4a*). A control experiment was performed to demonstrate that a brief reminder exposure of 5 min on its own was insufficient to induce new learning that formed a lasting memory (*Figure 4—figure supplement 1*). Mice given only a brief acquisition period of 5 min exhibited no preference for the novel object when tested 1 hr after training, suggesting the absence of a lasting object memory (*Figure 4—figure supplement 1*). We therefore used the 1 hr time point for the brief reminder experiment in *Figure 4a*. An AAV$_9$-TRE-ChR2-eYFP virus was injected into Fos-tTA mice to label engram cells in the dentate gyrus. During the object recall test, mice within the control group spent a similar amount of time exploring both the novel and familiar object, t(10) = 1.62, p=0.13, 95% CI [–1.13 to 7.22], again suggesting the original object memory was forgotten (*Figure 4b*). However, mice given a brief reminder session 1 hr prior to the recall test spent significantly more time exploring the novel object compared to the familiar object, t(4) = 4.63, p<0.01, 95% CI [7.27–29.00] (*Figure 4b*). In order to compare performance in object memory, a discrimination index was calculated. There was

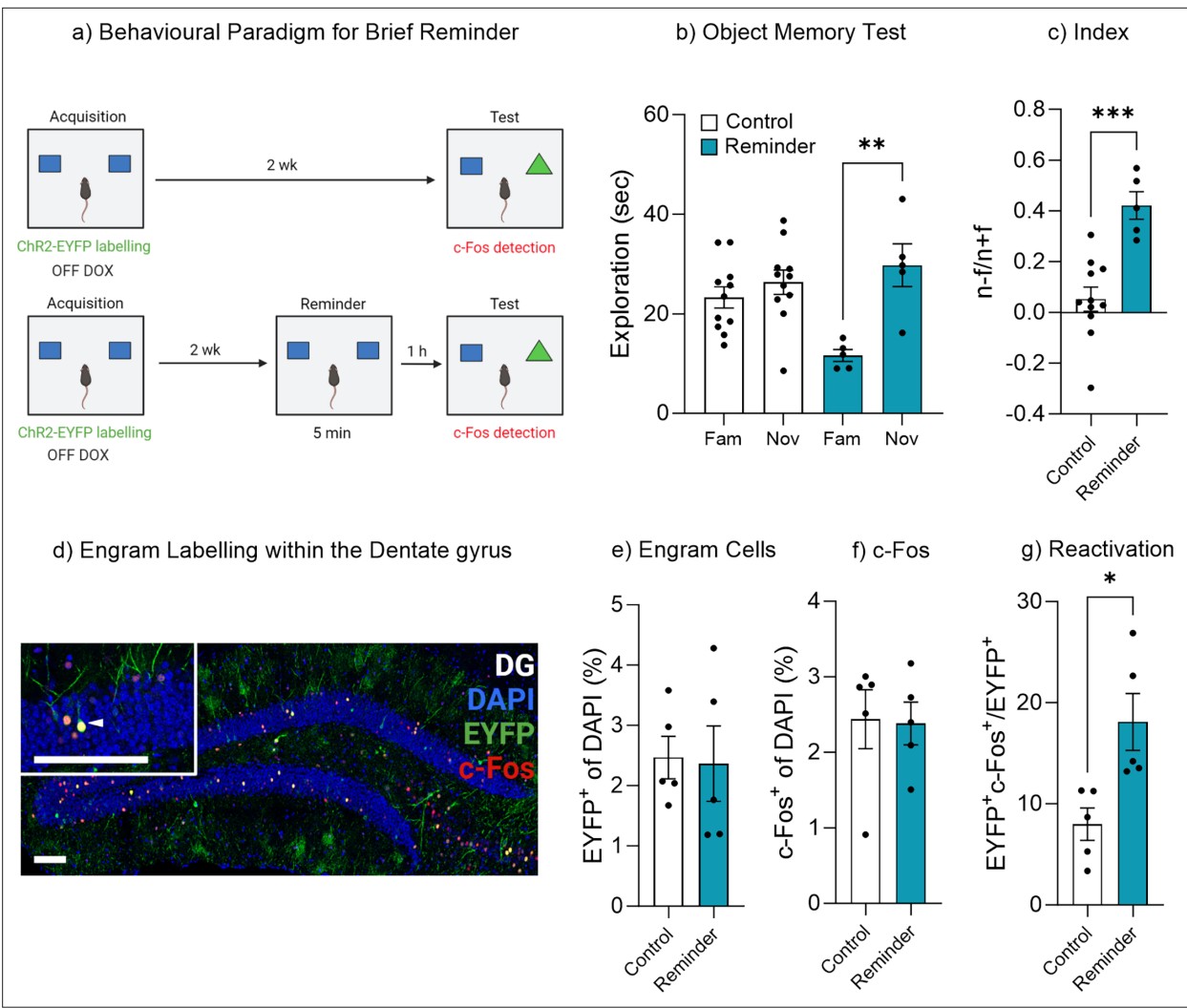

**Figure 4.** Exposure to the original stimuli facilities memory recall. (**a**) Behavioral timeline for a brief reminder exposure. (**b**) Object memory test. (**c**) Discrimination index. (**d**) Engram labeling within the dentate gyrus. (**e**) Engram cells. (**f**) c-Fos⁺ cells. (**g**) Engram reactivation. Bar graphs indicate average values in $n$=5–11 per group (*p<0.05, **p<0.01, ***p<0.001). Data graphed as means ± SEM. Scale bars 100 μm. Panel a was created with BioRender. com.

The online version of this article includes the following figure supplement(s) for figure 4:

**Figure supplement 1.** A single brief acquisition does not produce a lasting object memory.

**Figure supplement 2.** Repeated reminder exposure.

a significant difference between the discrimination index of the control and reminder group, t(14) = 4.57, p=0.001, 95% CI [0.19–0.54] (***Figure 4c***). Furthermore, the mice that underwent the reminder session and displayed intact memory recall also exhibited an increase in original engram reactivation, t(8) = 3.13, p=0.014, 95% CI [2.66–17.57] (***Figure 4d and g***), while there was no difference in the engram size, t(8) = 0.14, p=0.88, 95% CI [–1.76 to 1.55] (***Figure 4e***) or the level of neuronal activity within the dentate gyrus, t(8) = 0.11, p=0.91, 95% CI [–1.16 to 1.05] (***Figure 4f***). We next conducted an experiment to determine if repeated object reminder sessions would also reduce forgetting. In this paradigm, mice were repeatedly exposed to the original object pair (***Figure 4—figure supplement 2***). The results indicated that repeated reminder trials spaced over several days was also sufficient to facilitate persistent object memory recall (***Figure 4—figure supplement 2***). Together, these data indicate that a brief reminder or subsequent object reminders over time facilitates the transition of a forgotten memory to an accessible memory via modulating engram activity.

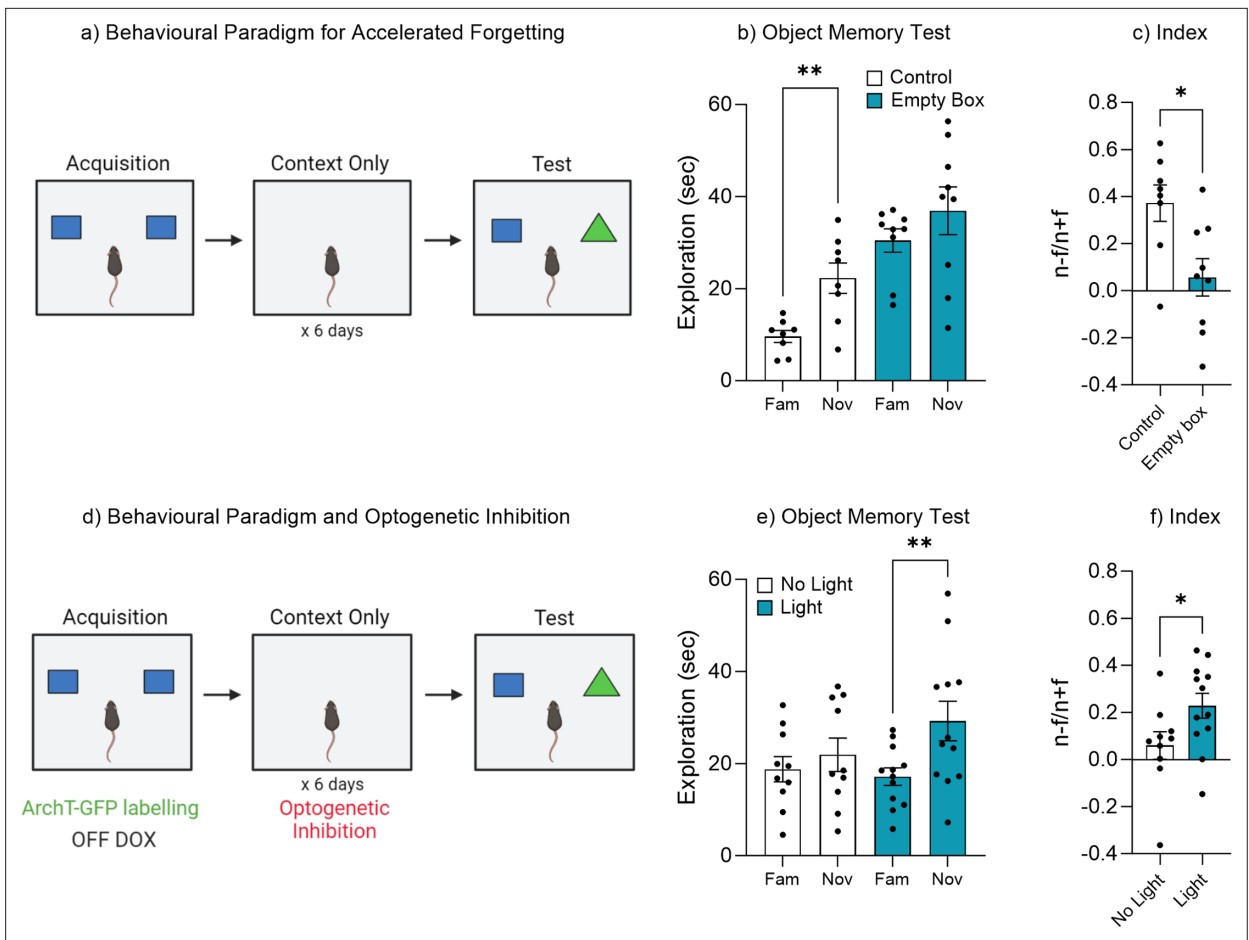

**Figure 5.** Repeated exposure to the training environment facilitates forgetting. (**a**) Behavioral paradigm. (**b**) Object memory test. (**c**) Discrimination index. (**d**) Behavioral paradigm for optogenetic inhibition during repeated context exposure. (**e**) Object memory test. (**f**) Discrimination index. Bar graphs indicate average values in $n$=8–12 per group (*p<0.05, **p<0.01). Data graphed as means ± SEM. Panel a was created with BioRender.com. Panel d was created with BioRender.com.

## Repeated context-only exposure to the training environment facilitates forgetting

Given that memory recall could be facilitated by optogenetic activation (**Figure 2**), maintained by enrichment (**Figure 3**), or even retrieved after forgetting by exposure to natural 'reminder' cues (**Figure 4**), we wanted to test that forgetting was driven by an adaptive process which updates memory engrams according to perceptual and environmental feedback. We therefore developed an altered version of the object memory task where mice were repeatedly reintroduced to the training context in the absence of objects (**Figure 5a**). We hypothesized that the repeated exposure to the training context without objects would update the original memory engram signaling that the objects were no longer relevant to the environment. During the recall test 1 week after training control mice spent significantly more time exploring the novel object compared to the familiar object, indicating a retrievable object memory, t(7) = 4.14, p=0.004, 95% CI [5.45–19.92] (**Figure 5b**). Whereas mice that had been repeatedly exposed to the training environment devoid of objects spent a similar amount of time exploring both the novel and familiar objects, t(8) = 1.26, p=0.24, 95% CI [–5.29 to 18.18], suggesting an updated and now inaccessible engram. In order to compare performance in object memory, a discrimination index was calculated. There was a significant difference in the discrimination index between the control and empty box exposed mice, t(15) = 2.83, p=0.012, 95% CI [–0.55 to –0.077] (**Figure 5c**). As the mice exhibit relatively high exploration of both the novel and familiar objects. An alternative explanation could be that the mice have not truly forgotten the familiar object, but rather as the mouse has not seen the familiar object in the previous six context-only sessions,

its reappearance makes it somewhat novel again. Therefore, this change in the object's reappearance might trigger the animal's curiosity, and in turn drives exploration by the animal. Indeed, cross-groups comparison reveals mice in the context-only group showed increased object exploration, F(3, 30)=11.30, p=0.005. A post hoc Tukey test showed that context-only mice explore the novel object, p=0.028, and object familiar, p=0.001 more than control mice. In order to confirm that the experience during the repeated exposure was indeed updating the original engram resulting in accelerated forgetting, we ran a second cohort of Fos-tTA mice which had been implanted with an optogenetic fiber and expressed the inhibitory opsin ArchT (*Figure 5d*). Following acquisition training the engram was optogenetically silenced during the repeated context-only exposures (*Figure 5d*). During the recall test, the No Light control group spent a similar amount of time exploring both the novel and familiar objects, t(9) = 1.36, p=0.20, 95% CI [–2.04 to 8.27] (*Figure 5e*), while the Light-induced inhibition mice, in which the engram was silenced, spent significantly more time exploring the novel object compared to the familiar object, t(11) = 3.60, p=0.004, 95% CI [4.70–19.46] (*Figure 5e*). In order to compare performance in object memory, a discrimination index was calculated. There was a significant difference in discrimination index between No Light control and Light-induced inhibition group, t(20) = 2.12, p=0.046, 95% CI [0.003–0.33] (*Figure 5f*). Combined these data suggest that experience following the encoding of an object memory can update the original memory engram accelerating the rate at which the information is forgotten. Further, this adaptive process requires the activity of the original engram ensemble.

## Rac1 mediates forgetting and engram reactivation

Rac1 has previously been shown to be involved in memory encoding and consolidation as well as active forgetting of contextual and social memory (*Davis and Zhong, 2017*; *Liu et al., 2018*; *Liu et al., 2016*; *Shuai et al., 2010*; *Cervantes-Sandoval et al., 2020*; *Berry et al., 2018*; *Wu et al., 2019*). Rac1 has also recently been shown to be involved in memory retrieval (*Lei et al., 2022*). We therefore sought to investigate the role of Rac1 signaling in mediating natural forgetting within the object memory paradigm employed in this study. An AAV$_9$-TRE-ChR2-eYFP virus was injected into Fos-tTA mice to label engram cells within the dentate gyrus. Immediately after acquisition training, mice were administered a Rac1 inhibitor (*Figure 6a*). Mice that were administered saline displayed typical forgetting at 2 weeks, as indicated by similar object exploration between the novel and familiar objects, t(8) = 1.07, p=0.31, 95% CI [–2.61 to 7.20] (*Figure 6b*), while mice treated with a Rac1 inhibitor displayed intact memory recall, spending significantly more time with the novel object compared to the familiar object, t(8) = 4.27, p=0.002, 95% CI [8.10–27.10] (*Figure 6b*). In order to compare performance in object memory, a discrimination index was calculated. There was a significant difference in discrimination index between the saline and Rac1 inhibitor group, t(17) = 3.53, p=0.002, 95% CI [0.12–0.49] (*Figure 6c*). Furthermore, there was no difference in engram size between the two groups, t(10) = 1.09, p=0.29, 95% CI [–0.45 to 1.33] (*Figure 6d*) or the level of neuronal activity within the dentate gyrus, t(10) = 0.86, p=0.40, 95% CI [–1.31 to 0.57] (*Figure 6e*). However, there was a significant decrease in engram reactivation, t(10) = 5.87, p<0.001, 95% CI [3.39–7.54] (*Figure 6f and g*). We also conducted a control experiment to demonstrate that Rac1 inhibition did not enhance memory recall at 24 hr (*Figure 6—figure supplement 1*).

Given Rac1 inhibition prevented forgetting, it stands to reason that by activating Rac1 forgetting may be accelerated. We next injected a Rac1 activator into a second cohort of Fos-tTA mice following memory encoding and again daily for 6 days (*Figure 6h*). During the object recall test, mice within the control group spent more time exploring the novel object compared to the familiar object, suggesting intact memory recall, t(8) = 4.98, p=0.001, 95% CI [9.11–24.78] (*Figure 6i*). Whereas mice with Rac1 activator spent a similar amount of time exploring both the novel and familiar object, suggesting the original object memory was forgotten, t(7) = 1.19, p=0.27, 95% CI [–2.33 to 7.05] (*Figure 6i*). In order to compare performance in object memory, a discrimination index was calculated. There was a significant difference between the discrimination index of the control and Rac1 activator group, t(15) = 3.68, p=0.002, 95% CI [–0.39 to –0.10] (*Figure 6j*). Furthermore, there was no difference in the engram size, t(12) = 0.29, p=0.77, 95% CI [–1.35 to 1.77] (*Figure 6k*) or the level of neuronal activity within the dentate gyrus, t(12) = 0.23, p=0.81, 95% CI [–1.09 to 1.36] (*Figure 6l*). However, there was a decrease in engram reactivation, t(12) = 3.15, p=0.008, 95% CI [–13.98 to –2.56] (*Figure 6m and n*). We also conducted a control experiment which indicated a single dose of Rac1 activator was sufficient

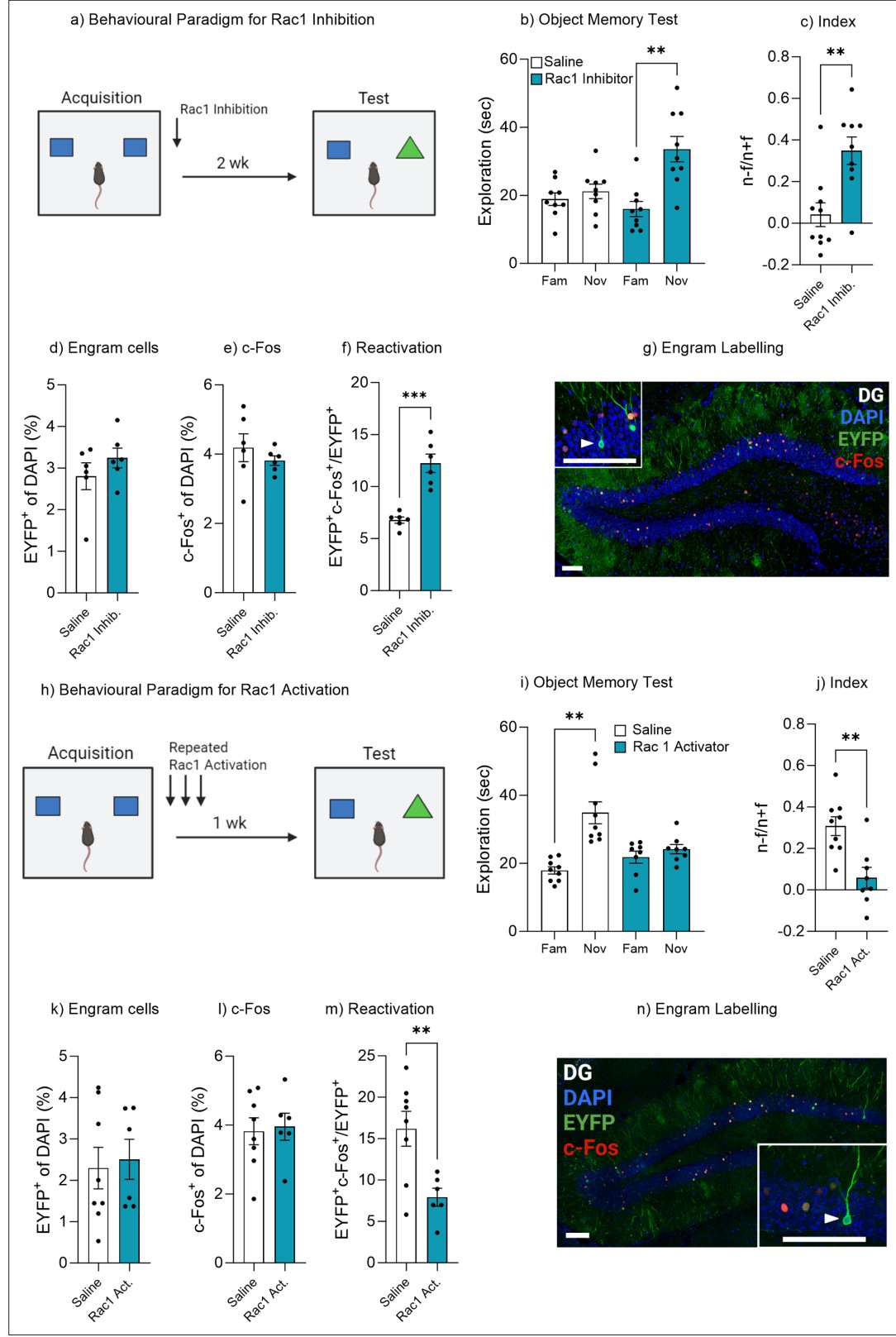

**Figure 6.** Rac1 modulates forgetting through engram activation. (**a**) Experimental timeline and drug administration for Rac1 inhibition. (**b**) Object memory test. (**c**) Discrimination index. (**d**) Engram cells. (**e**) c-Fos⁺ cells. (**f**) Engram reactivation. (**h**) Engram labeling within the dentate gyrus. (**i**) Experimental timeline and drug administration for Rac1 inhibition. (**j**) Object memory test. (**k**) Discrimination index. (**l**) Engram cells. (**m**) c-Fos⁺ cells. (**n**) Engram

*Figure 6 continued on next page*

*Figure 6 continued*

reactivation. Bar graphs indicate average values in *n*=8–9 per group (**p<0.01, ***p<0.001). Data graphed as means ± SEM. Scale bar 100 µm. Panel a was created with BioRender.com. Panel h was created with BioRender.com.

The online version of this article includes the following figure supplement(s) for figure 6:

**Figure supplement 1.** Rac1 inhibition prevents forgetting but does not enhance recognition memory.

**Figure supplement 2.** Rac1 activation accelerates forgetting.

to promote forgetting (*Figure 6—figure supplement 2*). Combined these data demonstrate the Rac1 signaling is involved in memory encoding and consolidation and may function as a plasticity enabling molecule to mediate retrievability. Rac1 may therefore be a signaling mechanism involved in adaptive forgetting.

## Learning model explains forgetting dynamics

Utilizing a mouse model of object memory, we have investigated the conditions in which a memory could be preserved, retrieved, or forgotten. The data indicate that forgetting is a dynamic process that can be explained as a gradient of engram activity, where the level of memory expression is experience-dependent and influenced by the animal's rate at which it learns from new experiences. Based on these findings, we sought to utilize the Rescorla-Wagner model to formalize the idea that forgetting is a form of learning and to integrate the empirical findings into a coherent, mechanistic, and quantitative framework (*Ryan and Frankland, 2022*; *Wagner and Rescorla, 1972*; *Todd and Holmes, 2022*). Our proposed Rescorla-Wagner model dynamically updates object-context associations in response to prediction errors (*Figure 7a*). Accordingly, engrams that are subjectively less relevant for adaptive behavior (representing weaker object-context associations) are more likely to be forgotten (*Todd and Holmes, 2022*). As illustrated in *Figure 7b*, 'engram strength' modeling the current object-context association is a function of the prediction error (difference between experienced objects and expected engram strength) and the 'learning rate' governing the influence of prediction errors on learning and forgetting. Positive prediction errors strengthen the engram, and negative errors yield weaker object-context associations (*Figure 7d*). Therefore, faster forgetting is driven by higher rates at which mice learn from negative prediction errors. In summary, this mechanism could help the animal prioritize relevant information in memory (*Figure 7c*). The animal predicts what it will encounter in its environment (e.g. what objects) and adjusts these predictions as a function of its experiences (e.g. the presence or absence of objects). Within this conceptual framework, the animal learns what features of the environment are currently relevant and thus important for remembering. In order to further support a link between our empirical data and computational modeling, we demonstrated that the modulation of engram cells within the dentate gyrus can indeed regulate object-context associations (*Figure 7—figure supplement 1*).

Supporting the idea that forgetting can be cast as an adaptive learning process, model simulations showed similar forgetting curves as mice in our experiment. We first applied the Rescorla-Wagner model to exploration behavior of mice in the standard housing condition that included data across four test time points (*Figure 3*). To obtain an empirical estimate of the subjects' learning rate governing the speed of forgetting (i.e. how quickly the object-context association degrades over time due to inhibitory object-context associations), we fitted the model to the data using a maximum-likelihood approach (see Materials and methods). Here, the learning rate indicating the weight given to negative prediction errors for updating engram strength was $\alpha^- = 0.07$, explaining the increase in forgetting across time (see *Supplementary file 1* for the other parameter estimates). Simulated forgetting curves that were based on the estimated parameters showed a close similarity to mice in the standard housing condition (*Figure 7e*; see *Figure 7—figure supplements 2 and 3* for a direct comparison of model predictions and empirical data).

The model also offers an explanation for the key results of the different experimental interventions. Experiencing objects during environmental enrichment that resemble the familiar objects in the memory test might nudge subjects to infer a higher engram relevancy that is more robust against prediction errors leading to higher engram expression (*Figure 8a, b, d, and f*). Supporting the perspective of forgetting as learning, the estimated learning rate in response to negative errors driving

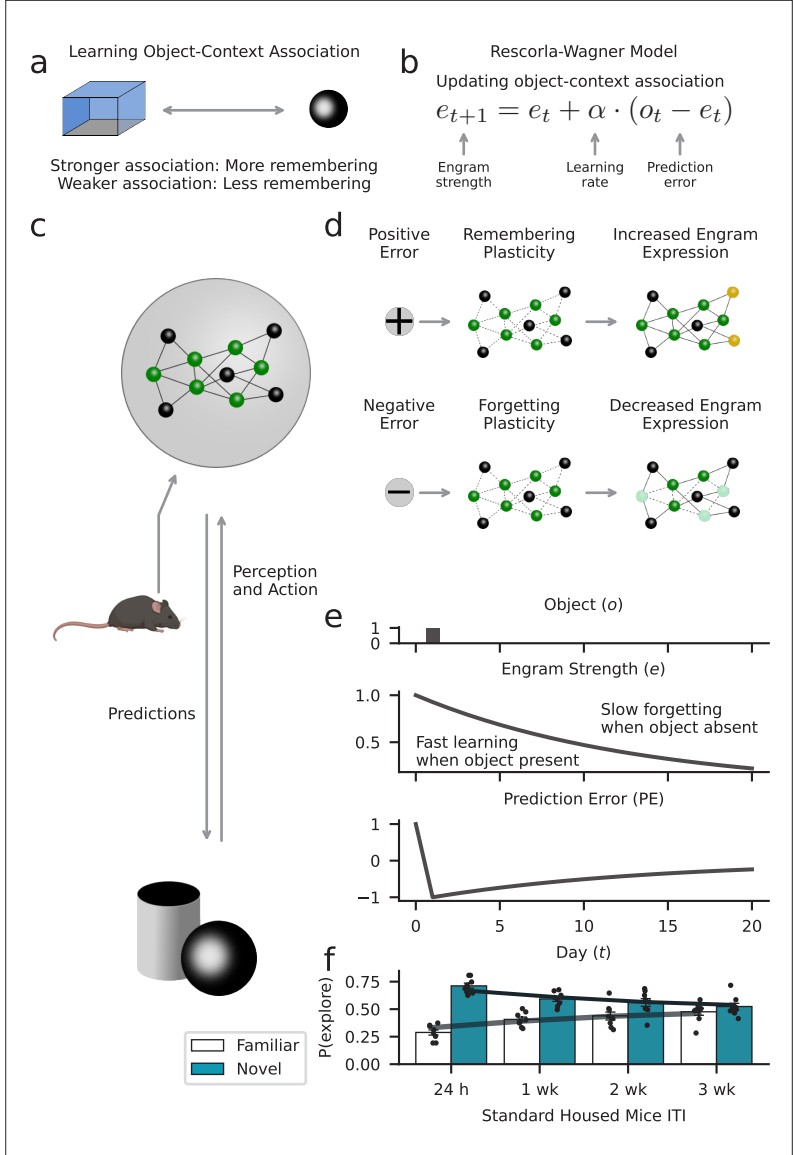

**Figure 7.** Forgetting as adaptive learning. Our model assumes that animals create and update memory engrams to flexibly adjust their behavior to their environment. (**a**) We used a Rescorla-Wagner model that learns object-context associations in the object-based memory task. The key assumption is that stronger associations lead to higher memory performance, while weaker associations lead to more forgetting. (**b**) The Rescorla-Wagner model updated the strength of object-context associations ('engram strength') as a function of the prediction error (difference between experienced object and engram strength) and the learning rates governing the influence of the prediction error. (**c**) Based on learned representations, animals constantly predict what happens in the environment (e.g. the occurrence of objects), and if predictions are violated (prediction errors), engrams are updated to improve the accuracy of future predictions. Here, established engram cells are shown in green; non-engram cells in gray. (**d**) Positive prediction errors signaling the occurrence of an unexpected event (e.g. new object) induce a learning process that increases the probability of remembering. This might rely on the recruitment of new engram cells (shown in yellow). In contrast, negative prediction errors signaling the absence of an expected event (e.g. predicted object did not appear) induce forgetting. This might rely on 'forgetting' plasticity reducing access to engrams (light green cells). (**e**) Our model formalizes this perspective based on the notion of 'engram relevancy', i.e., the strength of the object-context association. Higher engram relevancy makes it more likely that an engram is behaviorally expressed, e.g., through exploration behavior. The presentation of a novel object (upper panel) leads to a high engram relevancy (middle panel) in response to a positive prediction error (lower panel). The absence of an expected object decreases engram relevancy through negative prediction errors. (**f**) Model

*Figure 7 continued*

simulations corroborate the behavioral effects of our data (*Figure 3a*). Gray lines and bars show the average exploration probability for the familiar and novel object according to the model; markers show simulated mice.

The online version of this article includes the following figure supplement(s) for figure 7:

**Figure supplement 1.** Dentate gyrus engram modulates object context associations.

**Figure supplement 2.** In-sample model validation.

**Figure supplement 3.** Out-of-sample model validation.

---

forgetting $\alpha^- = 0$. Thereby, negative prediction errors did not induce forgetting so that memory performance was constant across 4 weeks (*Figure 7—figure supplement 2*). From a computational perspective, a low learning rate may also explain the high memory performance of the Rac1-inhibition group. For this group, model fitting revealed a similar learning rate to the enrichment group (learning rate for negative prediction errors $\alpha^- = 0.01$), suggesting that the inhibition of Rac1 made subjects largely ignore negative prediction errors that would normally drive forgetting, thereby avoiding the formation of inhibitory object-context associations. In contrast to reduced learning preventing forgetting due to Rac1 inhibition, Rac1 activation might speed forgetting through a higher learning rate. Similarly, we found accelerated forgetting following repeated exposure to the context-only training environment. Model simulations (*Figure 8c*) assuming a higher learning rate and, therefore, reduced engram expression, reproduced these findings (*Figures 5 and 6*).

A crucial assumption of our model is that seemingly forgotten engrams are not necessarily lost but rather inaccessible when the inferred object relevance is low. This is consistent with the experimental results above, where we presented evidence for the reinstatement of memory representations

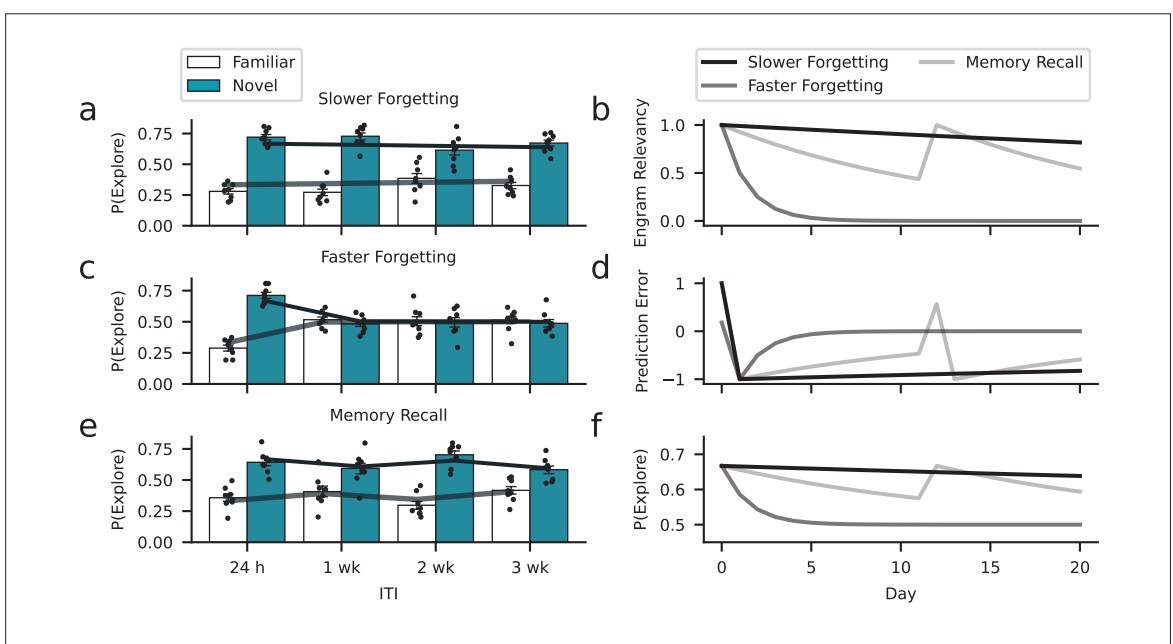

**Figure 8.** Learning model captures the dynamics of forgetting. Environmental, optogenetic, and pharmacological manipulations might modulate the speed of forgetting by altering key parameters of our model. Simulations with different learning-rate parameters explain the forgetting dynamics of the different experimental conditions. (**a**) The enrichment and Rac1-inhibition conditions were successfully captured using a low learning rate (0.01, similar to the empirical estimates). (**c**) In contrast, assuming a larger learning rate (0.5), we could capture faster forgetting as observed in the Rac1 activator and context-only conditions. (**e**) Moreover, improved memory performance after reminder cues can be explained by assuming that these interventions induce a positive prediction error boosting object relevancy. Here, we assumed a learning rate of 0.07 (based on the empirical estimate). (**b**) Development of engram relevancy and (**d**) prediction errors across conditions. (**f**) Probability of exploring the novel object plotted separately for each condition.

The online version of this article includes the following figure supplement(s) for figure 8:

**Figure supplement 1.** Model comparison.

**Figure supplement 2.** Parameter recovery.

based on real or artificial reminder cues (*Figures 2 and 4*). Here, we modeled memory reinstatement through reminder cues by inducing an artificial, positive prediction error in response to the reminder that boosted the inferred object relevance (strengthening object-context associations). Similar to the experimental data, the modeled reminder cues increased engram expression and yielded reduced forgetting rates (*Figure 8e*).

Finally, the model fits the data better than a baseline model predicting random forgetting rates, suggesting that it describes the data accurately and above chance level (*Figure 8—figure supplement 1*). We also performed a parameter-recovery analysis suggesting that parameters can sufficiently be estimated despite the limited data for model fitting (*Figure 8—figure supplement 2*). Our modeling results support the perspective that natural forgetting is a form of adaptive learning that alters engram accessibility in response to environmental feedback. Accordingly, environmental, optogenetic, and pharmacological manipulations might change the learning rate, suggesting that miscalibrated learning rates could indeed give rise to pathological forgetting. In summary, our Rescorla-Wagner model provides a mechanistic interpretation of our own data, a parsimonious explanation for the dynamics of natural forgetting, and could inform future studies on pathological forgetting.

## Discussion

For an organism to adapt to its environment, there must be processes by which learned information can be updated (*Pignatelli et al., 2019*; *Ryan and Frankland, 2022*; *Yassa and Stark, 2011*). Thus forgetting, or the letting go of information when it is appropriate, may be a key characteristic underlying learning and memory. It is possible that forgetting may be an adaptive process that is driven by feedback from the environment (*Ryan and Frankland, 2022*). This experience-dependent process could help an animal to prioritize relevant information in memory. Previous work has demonstrated that hippocampal engram cells are differently modulated by experience (*Lei et al., 2022*; *Denny et al., 2014*).

Here, we investigated the conditions in which a memory could be preserved, retrieved, or forgotten through changes in experience and learning. Altered memory recall was associated with corresponding changes in engram activation, indicating that altered memory engram expression is a reversible feature of natural forgetting. Under baseline conditions mice were able to recall an object memory 24 hr post-training but exhibited impaired recall after 2 weeks, indicating the initial memory was forgotten. The results demonstrated that engram activity correlated with the rate of forgetting. Moreover, these reactivated cells were shown to be a functional component of the initially encoded memory as the direct modulation of engram activity via optogenetic stimulation or inhibition either facilitated or prevented the recall of the object memory. These findings are in support of previous work which demonstrated hippocampal engram cells could be differently modulated by experience (*Denny et al., 2014*). These findings are also in agreement with the current literature which has shown that engram activation is associated with the level of memory recall (*Pignatelli et al., 2019*; *Finkelstein et al., 2022*; *Denny et al., 2014*; *Chen et al., 2019*; *Lacagnina et al., 2019*; *Gulmez Karaca et al., 2021*). In addition, they also support a role for the hippocampus as a fundamental structure for object memory (*Cohen and Stackman, 2015*; *Cohen et al., 2013*). It is important to consider that forgetting may also occur when an experience or an engram is not properly encoded during the learning phase, or not properly consolidated (*McGaugh, 2000*; *Dudai et al., 2015*; *Osorio-Gómez et al., 2019*). In the present study forgetting was also associated with a reduction in spine density relative to a 24 hr recall time point. This observation suggested that the memory was properly consolidated 1 day post-training (*Tonegawa et al., 2018*). We also observed intact memory recall at 1 week post-training, which further supports the idea that the initial object memory was indeed properly encoded and consolidated. Forgetting may encompass a broad spectrum of types of memory loss or apparent memory loss, some of which may involve the actual loss or damage of the engram, while others may reflect suppressed engram expression or simply altered conditions of engram activation. Here, we investigated spine density within engram cells following recall at 24 hr or 2 weeks. Future work could investigate if spine density or spine type is reduced in reactivated and non-reactivated engram cells within the ensemble and across engram and non-engram cells following different learning and forgetting conditions. This avenue of research could provide evidence to whether there are functionally distinct aspects of engram cells that are altered following forgetting (*Mishra et al., 2022*; *Kastellakis et al., 2023*).

Our data showing memory retrieval following optogenetic reactivation demonstrate that the forgotten engram is indeed still intact, otherwise activation of those cells would no longer trigger memory recall. Interestingly, previous work has shown that an engram remains intact even when memory consolidation is disrupted by protein synthesis inhibition or transgenic induction of Alzheimer's disease (*Perusini et al., 2017*; *Roy et al., 2016*; *Ryan et al., 2015*). In those studies, the engram persists in a latent state despite lower dendritic spine density (*Perusini et al., 2017*; *Roy et al., 2016*; *Ryan et al., 2015*). Moreover, optogenetic stimulation of amnesic engram cells was also sufficient to restore access to the amnesic memory (*Perusini et al., 2017*; *Roy et al., 2016*; *Ryan et al., 2015*). Taken together, these studies suggest the engram remains intact, while not properly retrievable (*Perusini et al., 2017*; *Roy et al., 2016*; *Ryan et al., 2015*).

We showed that the rate of forgetting and the level of engram activation could be modulated through enrichment. This enhanced memory recall was associated with increased engram reactivation. These findings are supported by earlier work, which showed exposure to an enriched environment prior to learning improved engram reactivation and rescued memory recall in a mouse model of Fragile X syndrome (*Li et al., 2020*). Previous literature has also shown the benefits of environmental enrichment in alleviating other pathological forms of memory impairment, such as Alzheimer's disease (*Barak et al., 2013*; *Griñán-Ferré et al., 2018*; *Jankowsky et al., 2005*). The effects of enrichment on learning and neural plasticity are well established and it is likely the contributing factor to the enhanced memory recall (*Bekinschtein et al., 2011*; *Clemenson et al., 2015*; *O'Leary et al., 2019a*; *Griñán-Ferré et al., 2018*; *Epp et al., 2021*; *Snyder and Drew, 2020*). A key component of enrichment is the increased perceptual and tactile feedback provided by extra items such as toys, objects, and running wheels, within the home cage for the mice to interact with and explore (*Sztainberg and Chen, 2010*; *Wolfer et al., 2004*). The added opportunity to experience these extra cage items, at an algorithmic level, may help to reinforce the original object memory, resulting in the preservation of an object's memory through multiple engram retrievals, and updating plasticity. Under this framework, the additional reminding experience of enrichment toys might help maintain or nudge mice to infer a higher engram relevancy that is more robust against forgetting, leading to higher engram expression. However, this line of investigation requires further empirical evidence to separate the impact of similar percepts on object memory recall from the broadly cognitive enhancing effects of environmental enrichment. Based on the computational analyses using the Rescorla-Wagner model we suggest that this form of natural forgetting is governed by the strength of the object-context associations, referred to here as engram relevancy. At the algorithmic level, stronger associations signal that the memory or engram representation is important or relevant and should not be forgotten. Accordingly, due to the experiences with extra objects in the enriched environment, mice might learn that objects are common in their environment and potentially relevant in the future. Thus, the object-context association is strong, preventing forgetting. It is important to emphasize that this process of updating is not necessarily separate from enrichment-induced plasticity at an implementational level, but part of the learning experience within an environment containing multiple objects. The enrichment or, more generally, experience, may therefore enhance memory through the modification of activity of specific engram ensembles. The idea of enrichment facilitating memory updating is consistent with the results obtained by the reminder experiments and further supported by our analysis with the Rescorla-Wagner computational model, where experience updates the accessibility of existing memories, possibly through reactivation of the original engram ensemble.

We validated the effectiveness of the enrichment paradigm to enhance neural plasticity by measuring adult hippocampal neurogenesis. The hippocampus has been identified as one of the only regions where postnatal neurogenesis continues throughout life (*Snyder and Drew, 2020*). Levels of adult hippocampal neurogenesis do not remain constant and can be altered by experience (*Clemenson et al., 2015*; *O'Leary et al., 2019a*; *O'Leary et al., 2019b*; *Denny et al., 2014*). In addition, adult-born neurons have also been shown to contribute to the process of forgetting (*Epp et al., 2021*; *Frankland et al., 2013*; *Akers et al., 2014*). Although the functional contribution of adult-born neurons to cognition and memory engrams is not fully understood (*Ko and Frankland, 2021*; *Tran et al., 2022*). *Mishra et al., 2022* showed that immature neurons were actively recruited into the engram following a hippocampal-dependent task. Increasing the level of neurogenesis rescued memory deficits by restoring engram activity, in a mouse model of Alzheimer's disease (*Mishra et al., 2022*). Specifically, augmenting neurogenesis rescued the deficits in spine density in both immature

and mature engram neurons (*Mishra et al., 2022*). Whether neurogenesis alters spine density differentially for reactivated or non-reactivation engrams cells remains to be investigated (*Mishra et al., 2022*; *Kastellakis et al., 2023*). This avenue of research would help to illuminate the morphological changes following forgetting and provide evidence if there is a functionally distinct aspect of engram cells that is altered in forgetting (*Mishra et al., 2022*; *Kastellakis et al., 2023*). Future work could utilize a different engram preparation based on a stable genetic labeling strategy could investigate the contribution of new-born neurons to the engram ensemble.

Our results showed that forgetting could be reversed following a brief reminder experience, which was associated with a corresponding increase in engram activity. Previous work from both human and rodent behavioral studies have shown that a brief exposure to reminder cues can aid memory recall (*Wahlheim et al., 2019*; *Tambini et al., 2017*; *Finkelstein et al., 2022*; *Bouton, 1993*). We further demonstrated that forgetting could be accelerated following repeated exposure to the training environment, therefore signaling the updated irrelevance of the object memory. This updating through experience was impaired when the original memory was inhibited. In line with the Rescorla-Wagner computational model, this suggests that in the present study, when mice are repeatedly exposed to the training context, the association between the context and the objects is reduced, resulting in a disrupted ability of the context to activate the object engram. In fact, earlier work by Bouton and colleagues has demonstrated that contextual changes from the passage of time can lead to retrieval failure (*Bouton, 1993*; *Bouton et al., 1999*; *Rosas and Bouton, 1997*). Future work addressing how changes in internal and external contextual cues alter memory expression will expand our understanding forgetting processes of multiple memory engrams, and their interactions.

Our findings indicate that the degree of learning and memory specificity corresponded with engram activity (*Leake et al., 2021*). The learning rate may therefore offer insight into the mechanism of forgetting. Here, the learning rate was altered following environmental enrichment as well as following repeated context exposure, which in turn affected the rate of forgetting. Together, this suggests the possibility of an 'optimal' learning rate that yields environmentally adaptive, natural forgetting and that learning rates that are too high may be linked to amnesia and learning rates that are too low to some sort of hypermnesia.

These findings raise an important question regarding the interpretation of forgetting (*Rosiles et al., 2020*; *Riccio and Richardson, 1984*). Previous work has emphasized retrieval deficits as a key characteristic of memory impairment, supporting the idea that memory recall or accessibility may be driven by learning through perceptual feedback from the environment (*Miller, 2021*; *Ryan and Frankland, 2022*; *Richards and Frankland, 2017*; *Perusini et al., 2017*; *Guskjolen et al., 2018*; *Roy et al., 2016*; *Autore et al., 2023*). Within our behavioral paradigm, a lack of memory expression would still constitute forgetting due to the loss of learned behavioral response in the presence of natural retrieval cues. The changes in memory expression may therefore underlie the adaptive nature of forgetting. Here, we studied natural forgetting, and our data showing memory retrieval following optogenetic reactivation demonstrates that the engram endures even when not expressed.

The Rescorla-Wagner model can be used to describe forgetting as a process of learning, where experience alters the degree of association between retrieval cues and memory (*Todd and Holmes, 2022*; *Yau and McNally, 2023*). Based upon this idea, we proposed a parsimonious computational model that integrates our findings into a cognitively plausible framework in which memories more relevant for adaptive behavior are more likely to be accessible than memories representing irrelevant, outdated information. As such, our modeling analyses suggest that the different experimental manipulations altered the learning rate determining how rapidly engrams switch from accessible to inaccessible states in response to environmental feedback, such as the presence or absence of objects. This model suggests that an emergent property underlying forgetting is the prediction error. Within this framework, forgetting is a form of learning whereby experience drives changes in the learning rate of an organism, and the expectation of one's environment can alter the efficiency of recall through forgetting plasticity (*Ryan and Frankland, 2022*). It therefore follows that prediction errors may determine whether an engram is strengthened, leading to higher engram expression, or weakened, leading to lower engram expression (*Figure 7d*). Positive prediction errors indicate that an unexpected event has taken place (e.g. some unexpected object is present). We assume that positive errors induce plasticity processes that alter the engram and increase the likelihood of engram expression. In contrast, negative errors indicate that something that was expected has not

taken place (e.g. object absent). Consequently, negative errors induce forgetting plasticity and yield decreased engram expression.

Dopamine is known to play a role in prediction error and learning rates (*Cai et al., 2020*; *Kalisch et al., 2019*). Moreover, dopamine has been shown to differently regulate Rac1 to modulate behavioral plasticity (*Davis and Zhong, 2017*; *Tu et al., 2019*). Here, we showed that the inhibition of Rac1 prevented forgetting, while its activation following memory encoding accelerated the rate of forgetting. This finding is in agreement with previous work where Rac1 impaired memory recall by driving forgetting-induced plasticity (*Davis and Zhong, 2017*; *Liu et al., 2018*; *Liu et al., 2016*; *Shuai et al., 2010*; *Cervantes-Sandoval et al., 2020*; *Wu et al., 2019*). Rac1 has been shown to function during the memory encoding and consolidation phases, and thereby to contribute to active forgetting (*Davis and Zhong, 2017*; *Liu et al., 2018*; *Liu et al., 2016*; *Shuai et al., 2010*; *Cervantes-Sandoval et al., 2020*). Recently, Rac1 has also been shown to be involved in memory retrieval (*Lei et al., 2022*). *Lei et al., 2022* demonstrated that a social reward enabled latent engrams to switch to active engrams via suppression of Rac1 activity in CA1 neurons of the hippocampus. It was further demonstrated that this effect was bidirectional as a social stressor switched active memory engrams back to latent engrams (*Lei et al., 2022*). This social stress-induced forgetting was modulated through the activation of Rac1 signaling (*Lei et al., 2022*). Overall, the activation of Rac1 has been shown to accelerate the rate of forgetting, while its inhibition prevented forgetting (*Liu et al., 2016*). Here, our work advanced this knowledge by demonstrating that Rac1-induced changes in memory recall alter engram reactivation. These works support our data suggesting that Rac1 is involved in altering the rate of forgetting at multiple time intervals after learning, not just immediately after learning. Our broader view is that Rac1 may function like a ubiquitous plasticity enabling molecule that is involved in memory encoding, consolidation, forgetting, and retrieval, depending on the experimental conditions and behavioral settings (*Ryan and Frankland, 2022*; *Davis and Zhong, 2017*).

Our Rescorla-Wagner model could be a starting point for more comprehensive models that account for forgetting across different experimental paradigms. Recent work suggests that retroactive interference emerges from the interplay of multiple engrams competing for expression (*Autore et al., 2023*). Further, there is the possibility of an interplay or connection between memory relevance and context within the process of extinction (*Bouton, 2004*). *Lacagnina et al., 2019* demonstrated that extinction training suppressed the reactivation of a fear engram, while activating a second putative extinction ensemble. In another study, these extinction engram cells and reward cells were shown to be functionally interchangeable (*Zhang et al., 2020*). Moreover, in a study conducted by *Lay et al., 2023*, the balance between extinction and acquisition was disrupted by inhibiting the extinction recruited neurons in the central nucleus of the amygdala. These results suggested that decision making after extinction was governed by a balance between acquisition and extinction-specific ensembles (*Lay et al., 2023*).

Future models could explicitly incorporate multiple engrams and their competition, explaining a broader range of forgetting effects. Several approaches to modeling extinction (*Redish et al., 2007*; *Gershman et al., 2010*; *Maren et al., 2013*), memory interference (*Gershman et al., 2014*), or the creation and updating of motor memories (*Heald et al., 2021*) that more explicitly assumed multiple memory representations could inform such a generalized model of natural forgetting. Furthermore, future work could also include other variables in addition to prediction errors to the model framework to better capture the dynamic conditions of memory recall. These may include boundary conditions of destabilization and reconsolidation, the salience or strength of the memory, as well as the timing of retrieval cues or updating experience. Future models could focus on understanding the specific boundary conditions in which a memory becomes retrievable and the degree to which it is sufficiently destabilized and liable to updating and forgetting. The role of perceptual learning in memory retrieval and forgetting may also be an avenue for future investigation. Understanding how experience alters object familiarity versus object retrieval and its impact on learning would also help to develop better models of object memory and forgetting.

Learning and memory allow humans and animals to maintain and update predictions about future outcomes. While having access to a large number of memories is adaptive in vast environments, it is equally important to prioritize the accessibility of the most relevant memories and to forget outdated information. Suppressing stored information through natural forgetting might therefore promote adaptive behavior. However, forgetting too much (e.g. amnesia) or too little (hypermnesia)

under pathological conditions is maladaptive. Here, we utilized a mouse model of object memory and investigated the conditions in which a memory could be preserved, retrieved, or forgotten. Moreover, through pharmacological and behavioral interventions, we successfully prevented or accelerated forgetting. Based on these findings and our computational model in which memories subjectively less relevant to adaptive behavior are more likely to be forgotten, we conclude that natural forgetting may be considered a form of adaptive learning where engrams that are subjectively less relevant for behavior are more likely to be forgotten.

## Materials and methods

### Animals

All wild-type behavior was conducted with male C57BL/6J mice aged 7–12 weeks, bred in-house from Charles River breeding pairs. Fos-tTA mice were generated in-house by breeding TetTag mice with C57BL/6J mice and selecting offspring carrying the Fos-tTA (*Ramirez et al., 2015*). All mice used for the experiments were male between 8 and 12 weeks of age at the time of surgery and had been raised on food containing 40 mg/kg doxycycline for at least 5 days prior to surgery. All mice were randomly assigned to experimental groups after surgery. Fos-tTA mice remained on doxycycline diet for the duration of the experiments. Doxycycline was removed from the diet for 36 hr prior to object acquisition to allow for engram labeling, following completion of the object acquisition period Fos-tTA mice were placed back on the doxycycline diet.

All mice were grouped housed in standard housing conditions, with temperature 22±1°C, relative humidity 50%, and a 12:12 hr light-dark cycle (lights on 0730 hr) and had ad libitum access to food and water. All experiments were conducted in accordance with the European Directive 2010/63/EU and under authorization issued by the Health Products Regulatory Authority Ireland and approved by the Animal Research Ethics Committee of Trinity College Dublin.

### Virus-mediated gene expression

The recombinant AAV vectors used for viral production were AAV-TRE-ChR2-eYFP and AAV-TRE-ArchT-GFP. Plasmids were serotyped with AAV9 coat proteins and packaged commercially by Vigene. The recombinant AAV vectors injected with viral titers were $1×10^{13}$ genome copy (GC) ml$^{-1}$ for AAV$_9$-TRE-ChR2-eYFP and AAV$_9$-TRE-ArchT-GFP.

### Stereotactic viral injection and optic fiber implantation

Mice were anesthetized with Avertin 500 mg/kg and placed into a stereotactic frame (World Precision Instruments). Bilateral craniotomies were performed using 0.5 mm diameter drill and the virus was injected using a 10 ml Hamilton microsyringe filled with mineral oil. A microsyringe pump and its controller were used to maintain the speed of the injection. The needle was slowly lowered to the target site and remained for 5 min before the beginning of the injection. For engram labeling and activation studies (AAV$_9$-TRE-ChR2-eYFP) or inhibition studies (AAV$_9$-TRE-ArchT-GFP) (volume; 300 nl) virus was injected bilaterally into the DG using the coordinates AP: –2 mm, ML: ±1.35 mm, DV: –2 mm relative to Bregma at a rate of 60 nl/min, followed by a 10 min diffusion period. For optogenetic experiments, a Doric optical fiber (200 μm core diameter; Doric Lenses) was implanted above the injection site (–2 mm AP; ±1.35 mm ML; –1.85 mm DV). A layer of adhesive cement (C&B Metabond) was applied followed by dental cement to secure the optic fiber implant to the skull. Mice were given 1.5 mg/kg meloxicam as analgesic. Animals were allowed 2 weeks to recover before behavioral testing. For optogenetic experiments. Control No Light mice underwent the identical surgery procedure with virus and optic fiber implantation. However, no light was delivered to excite or inhibit the respective opsin during the experiment.

### Engram labeling strategy

The AAV$_9$-TRE-ChR2-eYFP virus was injected into the dentate gyrus of Fos-tTA mice, under the control of a c-Fos promoter (*Ramirez et al., 2013*; *Ramirez et al., 2015*). The immediate early gene *Fos* becomes upregulated following neural activity and as such the Fos-tTA transgene selectively expresses tTA in active cells (*Reijmers et al., 2007*; *Reijmers and Mayford, 2009*). The activity-dependent tTA

induces expression of ChR2-eYFP. In order to restrict activity-dependent labeling to a single learning experience, mice were maintained on diet containing doxycycline.

## Object recognition

The novel object recognition task was based on the protocol previously described by *Bevins and Besheer, 2006*. Mice were first habituated to the testing arena (30 cm L × 20 cm W × 30 cm H) for a 10 min exploration period on 2 consecutive days. On day 3, two identical objects were positioned on adjacent sides of the arena and each animal was introduced for a 10 min exploration period. Mice were then placed directly back into their home cages. After a 24 hr or 2-week inter-trial interval, one familiar object was replaced with a novel object, and each animal was introduced for a 5 min exploration period. Objects and locations were counterbalanced across groups. Object exploration was defined when the animal's nose came in contact with the object. The testing arena and objects were cleaned with a disinfectant, TriGene, between each animal and training session. Video recordings were made to allow for manual scoring by experimenter blinded to group. The object discrimination index was calculated as the time spent exploring the novel object minus the time spent exploring the familiar object divided by the total time spent exploring both objects (novel–familiar/novel+familiar).

## Environmental enrichment

Mice were placed in the enriched housing for 3 weeks prior to behavioral testing and remained in the housing for the duration of the experiment. The enriched environment consisted of a larger home cage (50 cm L × 20 cm W × 40 cm H). The cage was included a running wheel, tunnels, extra nesting material, as well as an assortment of Lego bricks. The configuration of tunnels, Lego bricks, and housing enrichment changed each week.

## Pharmacological modification of Rac1

Rac1 activator (CN04) was reconstituted and prepared in saline at 2 µg/ml solution. CN04 was administered via intraperitoneal injection at a dose of 0.0125 mg/kg immediately following completion of the acquisition training. Mice were administered CN04 once per day for the following 5 days, for a total of six injections. Rac1 inhibitor (Ehop016) was reconstituted and prepared in PBS with 1% Tween-80, 1% DMSO, and 30% PEG. Ehop016 was administered via intraperitoneal injection at a dose of 20 mg/kg immediately following completion of the acquisition training.

## Immunohistochemistry

On completion of the behavioral testing, mice were deeply anesthetized with sodium pentobarbital and then transcardially perfused with 4% paraformaldehyde. Brains were post-fixed for 24 hr in 4% paraformaldehyde, transferred to PBS, and stored at 4°C. Coronal sections through the DG were collected onto slides at 50 µm thickness in a 1:4 series. Coronal sections were immunostained for eYFP and c-Fos. Non-specific antibody binding was blocked using 10% normal goat serum (NGS) in a solution of PBS with 0.2% Triton X-100 and tissue sections were incubated with goat anti-eYFP (Anti-GFP chicken IgY fraction 1:1000, Invitrogen) and anti-c-Fos (Anti-c-Fos rabbit, 1:500, Synaptic System). Sections were then incubated in the appropriate Alexa Fluor secondary antibody (AF488 for eYFP and AF596 for c-Fos) and then with DAPI (1:1000; Sigma) to stain nuclei. Lastly, sections were washed, mounted, and coverslipped with anti-fade mounting media (Vectashield-DAPI). Images were obtained using a Leica SP8 gated STED confocal microscope at ×40 magnification.

To measure adult-born neurons, mice were administered BrdU (50 mg/kg) 3 weeks before behavioral testing. For BrdU/NeuN immunohistochemistry, hippocampal sections were washed, denatured in HCl (2 N) for 45 min at 37°C, and renatured in 0.1 M sodium tetraborate. Sections were then washed in PBS and blocked in 10% NGS (Sigma) diluted in 0.1% Triton X-100 PBS to prevent non-specific binding. Sections were incubated with rat anti-BrdU antibody (1:100, Abcam) in 1% NGS diluted in 0.1% Triton X-100 PBS, washed, and then incubated in Alexa Fluor secondary antibody (AF598 for BrdU and AF88 for NeuN) and then with DAPI (1:1000; Sigma) to stain nuclei. Lastly, sections were washed, mounted, and coverslipped with anti-fade mounting media (Vectashield-DAPI). To quantify the cell counts, images were obtained using a Leica SP8 gated STED confocal microscope at ×40 magnification. The number of engram cells (eYFP+ cells) and c-Fos+ cells were counted. These were divided by the total number of DAPI-positive cells to estimate the percentage of c-Fos-positive

cells and eYFP-positive cells in the DG (both dorsal and ventral blade). To quantify the reactivation of engram cells, the number of cells positive for both eYFP and c-Fos was divided by the number of eYFP+ cells (*Autore et al., 2023*). The reactivation value for each mouse was calculated by averaging the reactivation for each slice (4 × sections per mouse) (*Autore et al., 2023*). To measure immature neurons, the total number of DCX-positive cells were counted in the dentate gyrus and averaged for each section (4 × sections per mouse). Cells expressing BrdU/NeuN were manually counted through the DG. The area of the DG for each animal was obtained using ImageJ software (National Institutes of Health, USA) and data for BrdU/NeuN or DCX-positive cells were expressed as the number of cells per µm$^2$ of DG (*Hueston et al., 2018*).

## Morphological analysis

To obtain engram dendritic spine density, the dentate gyrus was imaged using the Leica SP8 gated STED confocal microscope and images were collected with the Leica Application Suite X (LasX) software. Z-stacks were taken bidirectional under a 40× lens. Dendritic spine analysis was carried out using the Imaris software (Oxford Instruments, Imaris v9.5). The dendrites of eYFP-positive cells were traced using a semiautomated neurofilament tracer tool and dendritic spines were individually highlighted and manually traced with the software. The image processing feature of Imaris was used to apply the Gaussian and Median filters to Z-stack images to remove background eYFP staining and allow for better resolution and visualization of dendritic fragments and associated spines. Following the labeling of spines on traced dendritic fragments, parameters for spine volume, spine head volume, and dendritic spine density (/10 µm) were collected.

## Statistical analyses

All data were analyzed using SPSS statistical software (SPSS, Chicago, IL, USA). Behavioral data and the number of ChR2-eYFP-positive and c-Fos cells were graphed as means ± SEM. Data were analyzed by Student's t-test or ANOVA where appropriate. An alpha level of 0.05 was used as a criterion for statistical significance, and probability levels were quoted for non-significance.

## Computational modeling

We used an error-driven learning rule based on the Rescorla-Wagner model (*Daw and Tobler, 2014*; *Glimcher and Fehr, 2013*). The model offers a mechanistic explanation for the key behavioral results of our experiments (*Figures 7 and 8*). In line with the idea of the Rescorla-Wagner model that learning is an error-driven updating process, the model computes the relevancy of an engram $e_t$ on each day $t$ of the experiment, indicating the strength of the current object-context association. The engram relevancy is updated as a function of the prediction error ($o_t - e_t$, where $o_t$ denotes the object observation). The degree to which the prediction error changes the computed engram relevancy is determined by the learning rate $\alpha_t$. To update the engram relevancy, the model relies on a two-speeded mechanism through which the relevancy quickly increases when an object is present (higher learning rate) and slowly decays when the object is absent (lower learning rate). In particular, the appearance of an object yields a positive prediction error and triggers the fast-updating process, while the absence of an object is associated with negative prediction errors and slow decay in the engram relevancy. Depending on the time between the acquisition phase and the retrieval test, the model predicts different levels of memory performance, where a longer time interval between acquisition and test is associated with less object recognition.

## Task model

To model mouse behavior in the object-based memory task, we first formulated a model of the experimental paradigm.

- $T := 21$ denotes the maximum number of days, which are indexed as $t = 0, 1, \ldots, T$.
- $x \in \{0, 1\}$ denotes the objects of the task. $x = 0$ refers to the novel object; $x = 1$ refers to the acquisition object that is presented together with the novel object.
- $O \in \{0, 1\}$ indicates whether an object is presented. When the acquisition (and therefore familiar) object $x = 1$ is presented in the acquisition phase, $o_t^{x=1} = 1$. Moreover, $o_t^{x=1} = 1$ when the retrieval test takes place, which depending on the task condition is after 1 day, 1 week, 2 weeks, or 3 weeks. In all other cases, $o_t^{x=1} = 0$.

## Learning model

To formalize the learning and memory processes of mice in the object-based memory task, we developed a simple reinforcement-learning model based on the Rescorla-Wagner model. Here,

- $e_t \in [0, 1]$ denotes the relevancy of an engram, which is updated on each trial. Higher values indicate higher object relevancy (stronger object-context association).

The key assumption of the model is that once engrams are formed, they endure but might be inaccessible when they cannot be reactivated following natural forgetting. The expressibility of the engram depends on the inferred relevancy of the respective object, where objects that are subjectively more likely to be encountered in the current environment (higher relevancy) are more easily accessible. The inferred relevancy of the (familiar) acquisition object $x = 1$ governing the accessibility of the engram was computed according to the delta rule

$$e_{t+1} = e_t + \alpha_t \left( o_t - e_t \right) \tag{1}$$

where $e_t$ denotes the relevancy of the engram, $\alpha_t$ the learning rate, and $o_t - e_t =: \delta_t$ the prediction error. For clarity, we omit the dependency on $x$ here. Moreover, we used separate learning rates for positive and negative prediction errors

$$\alpha_t := \begin{cases} 1, & \delta_t \geq 0 \\ \alpha^-, & \delta_t < 0 \end{cases} \tag{2}$$

That is, positive prediction errors indicating the presence of an object lead to a rapid increase in the relevancy of the engram. Negative prediction errors indicating the absence of an object lead to a slower decrease in the relevancy (when $\alpha^- < 1$).

## Exploration model

To translate the modeled engram relevancy into exploration behavior, we used the softmax and beta functions. The softmax function was used to compute the average exploration probability for the familiar object

$$\mu = \frac{\exp\left(\beta e_t^{x=1}\right)}{\sum_x \exp\left(\beta e_t^x\right)} \tag{3}$$

where the inverse temperature parameter $\beta$ determines the slope of the function. $e_t^{x=1}$ denotes the engram relevancy of the familiar object that is dynamically updated according to the delta rule (*Equation 1*). $e_t^{x=0}$ corresponds to the relevancy of the novel object, where, since the object has not been experienced before, we assume an engram relevancy of 0. Subsequently, we utilized the beta distribution to model exploration variability across individuals. The beta distribution for $\mu \in [0, 1]$ is defined by

$$Beta\left(\mu; a, b\right) := \frac{1}{B\left(a, b\right)} \mu^{a-1} \left(1 - \mu\right)^{b-1} \tag{4}$$

with the shape parameters $a$ and $b$. In order to compute the distribution over exploration probability conditional on the mean $\mu$, we exploited that the sum of $a$ and $b$ indicates the concentration $\kappa$ of the distribution. That is, when $\kappa = a + b$ gets larger, the beta distribution is more concentrated, and the predicted exploration variability is lower. When $\kappa$ is lower, the distribution is wider and the model predicts greater exploration variability (for more details, see *Krushke, 2014*).

Therefore, to compute the exploration probabilities for object $x = 1$ using the beta distribution conditional on the computed average exploration probability $\mu$ and concentration $\kappa$, we computed the shape parameters $a$ and $b$ in the following way:

$$a = \mu\kappa \text{ and } b = \left(1 - \mu\right)\kappa. \tag{5}$$

## Parameter estimation

We implemented the following free parameters. The learning rate conditional on negative prediction errors $\alpha^-$ (see *Equation 2*), the inverse temperature parameter $\beta$ of the softmax distribution (see *Equation 3*), and the $\kappa$ parameter of the beta distribution (see *Equations 4 and 5*). To estimate these parameters, we computed the probability of the observed exploration probabilities $\mu$ for the familiar object $x = 1$ conditional on $a$ and $b$ on the test day using the beta distribution (*Equation 4*). Based upon this, we computed the model fit across subjects of the experimental and control group by summing the log-likelihoods

$$\mathfrak{l} = \sum_s \log p\left(\hat{\mu}_s | a_s, b_s\right) \tag{6}$$

where $s$ denotes the subject. The free parameters were estimated using the bound-constrained optimization algorithm L-BFGS-B of the SciPy library in Python 3.10. The parameter boundaries were [0, 0.25] for $\alpha^-$, [–10, 0] for $\beta$, and [1, 100] for $\kappa$.

## Model comparison

We systematically compared our model to a baseline model that explored the two objects with the same probability ($\mu = 0.5$), where we computed the log-likelihood based on $a = 1$ and $b = 1$. We compared the Bayesian information criterion (BIC; *Stephan et al., 2009*) of the two models defined by

$$BIC := \ell - \frac{k}{2} \log S \tag{7}$$

where $k$ denotes the number of free parameters and $S$ the number of subjects of the respective group.

## Acknowledgements

We thank Tamara Boto for many useful discussions and Eric Patrick Byrne for assistance with engram morphology analysis. We also thank Andrea Muñoz Zamora and Aaron Douglas for proofreading and critical feedback on the manuscript, as well as past and present Ryan Lab members for collegial support and scientific input. This work was funded by Science Foundation Ireland (SFI), the European Research Council (ERC), the Irish Research Council (IRC), the US Air Force Office of Scientific Research (AFOSR), and the Jacobs Foundation. RB was supported by Deutsche Forschungsgemeinschaft (DFG). Images for the experimental timelines were produced using BioRender.

## Additional information

### Competing interests

Tomás J Ryan: Reviewing editor, *eLife*. The other authors declare that no competing interests exist.

### Funding

| Funder | Grant reference number | Author |
| --- | --- | --- |
| Science Foundation Ireland | 15/YI/3187 | Tomás J Ryan |
| Irish Research Council | GOIPD/2019/813 | James D O'Leary Tomás J Ryan |
| European Research Council | 715968 | Tomás J Ryan |
| Air Force Office of Scientific Research | FA9550-20-1-0316 | Tomás J Ryan |
| Jacobs Foundation | Fellowship | Tomás J Ryan |
| Irish Research Council | GOIPG/2018/2357 | Livia Autore Tomás J Ryan |

| Funder | Grant reference number | Author |
|---|---|---|
| Deutsche Forschungsgemeinschaft | 412917403 | Rasmus Bruckner |

The funders had no role in study design, data collection and interpretation, or the decision to submit the work for publication.

## Author contributions

James D O'Leary, Conceptualization, Resources, Data curation, Formal analysis, Validation, Investigation, Visualization, Methodology, Writing - original draft, Project administration, Writing - review and editing; Rasmus Bruckner, Conceptualization, Resources, Data curation, Software, Formal analysis, Investigation, Visualization, Methodology, Writing - review and editing, Computational modelling; Livia Autore, Data curation, Formal analysis, Validation, Investigation, Methodology, Writing - review and editing; Tomás J Ryan, Conceptualization, Resources, Supervision, Funding acquisition, Visualization, Methodology, Writing - original draft, Project administration, Writing - review and editing

## Author ORCIDs

Rasmus Bruckner https://orcid.org/0000-0002-3033-6299
Tomás J Ryan https://orcid.org/0000-0003-0121-8514

## Ethics

All experiments were conducted in accordance with the European Directive 2010/63/EU and under an authorization issued by the Health Products Regulatory Authority (HPRA) in Ireland. All work was also approved by the Animal Research Ethics Committee of Trinity College Dublin.

Reviewer #1 (Public Review): https://doi.org/10.7554/eLife.92860.3.sa1
Reviewer #2 (Public Review): https://doi.org/10.7554/eLife.92860.3.sa2
Reviewer #3 (Public Review): https://doi.org/10.7554/eLife.92860.3.sa3
Author response https://doi.org/10.7554/eLife.92860.3.sa4

# Additional files

## Supplementary files

• Supplementary file 1. Parameter estimates. Control group (Cont) and experimental group (Exp).
• MDAR checklist

## Data availability

Source data for all Figures can be accessed here: https://osf.io/96z7t. Modeling code and behavioral data are available at https://github.com/rasmusbruckner/engram_eLife, (copy archived at *Bruckner, 2024*).

The following dataset was generated:

| Author(s) | Year | Dataset title | Dataset URL | Database and Identifier |
|---|---|---|---|---|
| James O | 2024 | Natural forgetting reversibly modulates engram expression | https://osf.io/96z7t | Open Science Framework, 96z7t |

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
