## [Editor Report · eLife assessment]

This is an **important** paper on the role of engrams and relevant conditions that influence memory and forgetting. The variety of methods used, namely, behavioural, labeling, interrogation, immunohistochemistry, microscopy, pharmacology, computational, are exemplary and provide **convincing** evidence for the role of engrams in the dentate gyrus in memory retrieval and forgetting. This examination will be of interest broadly across behavioural and neural science communities.

---

## [Referee Report · Reviewer #1 (Public Review)]

O'Leary and colleagues sought to understand the factors that underlie memory processes, including formation, retrieval, and forgetting. The present data identify time, environmental enrichment, Rac-1, context reexposure, and brief reminders of the familiar object as factors that alter discrimination between novel and familiar objects. This is complimented with an engram approach to quantify cells that are active during learning to examine how their activation is impacted with each of the above factors at test. There are many strengths in the manuscript, including systematic testing of several factors that contribute to poor discrimination between novel and familiar objects. These results are interesting and outline essential boundaries of incidental, nonaversive memory. With this behavioral data, authors apply a modeling approach to understand the factors that contribute to good and poor object memory recall.

---

## [Referee Report · Reviewer #2 (Public Review)]

Summary:

The manuscript examines an important question about how an inaccessible, natural forgotten memory can be retrieved through engram ensemble reactivation. It uses a variety of strategies including optogenetics, behavioral and pharmacological interventions to modulate engram accessibility. The data characterize the time course of natural forgetting using an object recognition task, in which animals can retrieve 1 day and 1 week after learning, but not 2 weeks later. Forgetting is correlated with lower levels of cell reactivation (c-fos expression during learning compared to retrieval) and reduction in spine density and volume in the engram cells. Artificial activation of the original engram was sufficient to induce recall of the forgotten object memory while artificial inhibition of the engram cells precluded memory retrieval. Mice housed in an enriched environment had a slower rate of forgetting, and a brief reminder before the retrieval session promoted retrieval of a forgotten memory. Repeated reintroduction to the training context in the absence of objects accelerated forgetting. Additionally, activation of Rac1-mediated plasticity mechanisms enhanced forgetting, while its inhibition prolonged memory retrieval. Authors also reproduce the behavioral findings using a computational model inspired by Rescorla-Wagner model. In essence, the model proposes that forgetting is a form of adaptive learning that can be updated based on prediction error rules in which engram relevancy is altered in response to environmental feedback.

Strengths:

(1) The data presented in the current paper are consistent with the authors claim that seemingly forgotten engrams are, in fact, accessible. This suggests that retrieval deficits can lead to memory impairments rather than a loss of the original engram (at least in some cases).

(2) The experimental procedures and statistics are appropriate, and the behavioral effects appear to be very robust. Several key effects are replicated multiple times in the manuscript.

Comments on revised version:

The authors have adequately addressed my prior concerns.

---

## [Referee Report · Reviewer #3 (Public Review)]

Summary:

The manuscript by Ryan and colleagues uses a well-established object recognition task to examine memory retrieval and forgetting. They show that memory retrieval requires activation of the acquisition engram in the dentate gyrus and failure to do so leads to forgetting. Using a variety of clever behavioural methods, the authors show that memories can be maintained and retrieval slowed when animals are reared in environmental enrichment and that normally retrieved memories can be forgotten if exposed to the environment in which the expected objects are no longer presented. Using a series of neural methods, the authors also show that activation or inhibition of the acquisition engram is key to memory expression and that forgetting is due to Rac1.

Strengths:

This is an exemplary examination of different conditions that affect successful retrieval vs forgetting of object memory. Furthermore, the computational modelling that captures in a formal way how certain parameters may influence memory provides an important and testable approach to understanding forgetting.

The use of the Rescorla-Wanger model in the context of object recognition and the idea of relevance being captured in negative prediction error are novel (but see below).

The use of gain and loss of function approaches are a considerable strength and the dissociable effects on behaviour eliminate the possibility of extraneous variables such as light artifacts as potential explanations for the effects.

Weaknesses:

A closer examination of the process that governs the behavioural effect in the present investigation would have been of even greater significance. The authors acknowledge the distinction between object familiarity vs object recognition, but a direct assessment would benefit the field's understanding of the current role of engrams on behaviour.

Relatedly, while relevance is an interesting concept that has been operationalized in the paper, it is unclear how distinct it is from extinction. Specifically, in the case where the animals are exposed to the context in the absence of the object, the paper currently expresses this as a process of relevance - the objects are no longer relevant in that context. Another way to think about this is in terms of extinction - the association between the context and the objects is reduced resulting in a disrupted ability of the context to activate the object engram. The authors have noted the potential role of extinction in their studies.

The impact of the paper is somewhat limited by the use of only one sex. Do the authors expect an identical process to be engaged in females in the present set of studies?

---

## [Author Response]

The following is the authors’ response to the original reviews.

**Public Reviews:**

**Reviewer #1 (Public Review):**
Summary:O'Leary and colleagues present data identifying several procedures that alter discrimination between novel and familiar objects, including time, environmental enrichment, Rac-1, context reexposure, and brief reminders of the familiar object. This is complimented with an engram approach to quantify cells that are active during learning to examine how their activation is impacted following each of the above procedures at test. With this behavioral data, authors apply a modeling approach to understand the factors that contribute to good and poor object memory recall.

We thank the Reviewer for summarizing the scope and depth of our manuscript, and indeed for recognizing our efforts. We engage below with the Reviewer’s specific criticisms.

Strengths:Authors systematically test several factors that contribute to poor discrimination between novel and familiar objects. These results are extremely interesting and outline essential boundaries of incidental, nonaversive memory.These results are further supported by engram-focused approaches to examine engram cells that are reactivated in states with poor and good object recognition recall.

We thank the Reviewer for these positive comments.

Weaknesses:For the environmental enrichment, authors seem to suggest objects in the homecage are similar to (or reminiscent of) the familiar object. Thus, the effect of improved memory may not be related to enrichment per se as much as it may be related to the preservation of an object's memory through multiple retrievals, not the enriching experiences of the environment itself. This would be consistent with the brief retrieval figure. Authors should include a more thorough discussion of this.

This is one of the main issues highlighted by the Editor and the Reviewers. We agree that these results dove-tail with the reminder experiments. We have included additional discussion, see line 510-546.

Authors should justify the marginally increased number of engram cells in the non-enrichment group that did not show object discrimination at test, especially relative to other figures. More specific cell counting criteria may be helpful for this. For example, was the DG region counted for engram and cfos cells or only a subsection?

There was a marginal, but non-significant increase in the number of labelled cells within the standard housed mice in Figure 3f. The cell counting criteria was the same across experimental groups and conditions, where the entire dorsal and ventral blade of the dorsal DG was counted for each animal. This non-statistically significant variance may be due to surgical and viral spread difference between mice. We have clarified this in the manuscript, see line 229-232.

It is unclear why the authors chose a reactivation time point of 1hr prior to testing. While this may be outside of the effective time window for pharmacological interference with reconsolidation for most compounds, it is not necessarily outside of the structural and functional neuronal changes accompanied by reconsolidation-related manipulations.

A control experiment was performed to demonstrated that a brief reminder exposure of 5 mins on its own was insufficient to induce new learning that formed a lasting memory (Supplementary Figure S4a). Mice given only a brief acquisition period of 5 mins, exhibited no preference for the novel object when tested 1 hour after training, suggesting the absence of a lasting object memory (Supplementary Figure S4b & c). We therefore used the 1-hour time point for the brief reminder experiment in Figure 4a. We have clarified this within the manuscript and supplementary data see line 258-264.

Figure 5: Levels of exploration at test are inconsistent between manipulations. This is problematic, as context-only reexposures seem to increase exploration for objects overall in a manner that I'm unsure resembles 'forgetting'. Instead, cross-group comparisons would likely reveal increased exploration time for familiar and novel objects. While I understand 'forgetting' may be accompanied by greater exploration towards objects, this is inconsistent across and within the same figure. Further, this effect is within the period of time that rodents should show intact recognition. Instead, context-only exposures may form a competing (empty context) memory for the familiar object in that particular context.

The Reviewer raises an important question, and we agree with the Reviewer that there should be caution and qualification around interpreting these results as “forgetting”. Indeed, for the context-only rexposures, cross-group comparisons show increased exploration time for familiar and novel objects. As the mice exhibit relatively high exploration of both the novel and familiar objects. An alternative explanation would be that the mice have not truly forgotten the familiar object, but rather as the mouse has not seen the familiar object in the last 6 context only sessions, its reappearance makes it somewhat novel again. Therefore, this change in the object’s reappearance triggers the animal’s curiosity, and in turn drives exploration by the animal. In addition, the context-only exposures may form a competing memory for the familiar object in that particular context. We have highlighted this in the results and also included greater discussion. See lines 306-315.

I am concerned at the interpretation that a memory is 'forgotten' across figures, especially considering the brief reminder experiments. Typically, if a reminder session can trigger the original memory or there is rapid reacquisition, then this implies there is some savings for the original content of the memory. For instance, multiple context retrievals in the absence of an object reminder may be more consistent with procedures that create a distinct memory and subsequently recruit a distinct engram.

These findings raise an important question regarding the interpretation of ‘forgetting’. If a reminder trial or experience can trigger the original memory, or there is rapid reacquisition, then this would suggest there is a degree of savings for the original memory content (85, 86). Previous work has emphasized retrieval deficits as a key characteristic of memory impairment, supporting the idea that memory recall or accessibility may be driven by learning feedback from the environment (7, 8, 14–18). Within our behavioral paradigm, a lack of memory expression would still constitute forgetting due to the loss of learned behavioral response in the presence of natural retrieval cues. The changes in memory expression may therefore underlie the adaptive nature of forgetting. This is consistent with the idea that the engram is intact and available, but not accessible. Here we studied natural forgetting, and our data showing memory retrieval following optogenetic reactivation demonstrates that the original engram persists at a cellular level, otherwise activation of those cells would no longer trigger memory recall. We also agree with the reviewer that multiple context retrievals may indeed lead to the formation of a second distinct engram that competes with the original. Recent work suggests that retroactive interference emerges from the interplay of multiple engrams competing for accessibility (18). We have added clarification and included extra discission of this interpretation. See lines 589-598.

Authors state that spine density decreases over time. While that may be generally true, there is no evidence that mature mushroom spines are altered or that this is consistent across figures. Additionally, it's unclear if spine volume is consistently reduced in reactivated and non-reactivated engram cells across groups. This would provide evidence that there is a functionally distinct aspect of engram cells that is altered consistently in procedures resulting in poor recognition memory (e.g. increased spine density relative to spine density of non-reactivated engram cells and non-engram cells)

We thank the Reviewer for their helpful comments on explaining our engram dendritic spine data. We agree with the Reviewer that an analysis of the changes in spine type, as well as the difference between engram and non-engram spines as well and reactivation and non-reactivated engram spines would be interesting and may help to further illuminate the morphological changes of forgetting and memory retrieval. Indeed, future analysis could determine if spine density is reduced in reactivated and non-reactivated engram cells or indeed across engram non-engram cells within different learning conditions. This avenue of investigation could determine if there is a functionally distinct aspect of engram cells that are altered following forgetting (67). However, such analysis is beyond the scope of this study. We have highlighted this limitation and included its discussion. See lines 493-499.

Authors should discuss how the enrichment-neurogenesis results here are compatible with other neurogenesis work that supports forgetting.

We validated the effectiveness of the enrichment paradigm to enhance neural plasticity by measuring adult hippocampal neurogenesis. The hippocampus has been identified as one of the only regions where postnatal neurogenesis continues throughout life (75). Levels of adult hippocampal neurogenesis do not remain constant throughout life and can be altered by experience (41–43, 57). In addition, adult born neurons have been shown to contribute to the process of forgetting (74, 78, 79). Although the contribution of adult born neurons to cognition and the memory engram is not fully understood (80, 81). Mishra et al, showed that immature neurons were actively recruited into the engram following a hippocampal-dependent task (67). Moreover, increasing the level of neurogenesis rescued memory deficits by restoring engram activity (67). Augmenting neurogenesis further rescued the deficits in spine density in both immature and mature engram neurons in a mouse model of Alzheimer’s disease (67). Whether neurogenesis alters spine density on differentially for reactivated or non-reactivation engrams cells remains to be investigated (67, 68). This avenue of research would help to illuminate the morphological changes following forgetting and provide evidence if there is a functionally distinct aspect of engram cells that is altered in forgetting (67, 68). Our engram labelling strategy which utilized c-fos-tTA transgenic mice combined with an AAV9-TRE-ChR2-eYFP virus does not necessarily label sufficient immature neurons. Future work could utilize a different engram preparation, such as a genetic labelling strategy (TRAP2) or using a different immediate early gene promoter such as Arc to investigate the contribution of new-born neurons to the engram ensemble. We have added additional discussion of how our work fits with previous literature investigating neurogenesis and forgetting. See lines 547-565.

**Reviewer #2 (Public Review):**
Summary:The manuscript examines an important question about how an inaccessible, natural forgotten memory can be retrieved through engram ensemble reactivation. It uses a variety of strategies including optogenetics, behavioral and pharmacological interventions to modulate engram accessibility. The data characterize the time course of natural forgetting using an object recognition task, in which animals can retrieve 1 day and 1 week after learning, but not 2 weeks later. Forgetting is correlated with lower levels of cell reactivation (c-fos expression during learning compared to retrieval) and reduction in spine density and volume in the engram cells. Artificial activation of the original engram was sufficient to induce recall of the forgotten object memory while artificial inhibition of the engram cells precluded memory retrieval. Mice housed in an enriched environment had a slower rate of forgetting, and a brief reminder before the retrieval session promoted retrieval of a forgotten memory. Repeated reintroduction to the training context in the absence of objects accelerated forgetting. Additionally, activation of Rac1-mediated plasticity mechanisms enhanced forgetting, while its inhibition prolonged memory retrieval. The authors also reproduce the behavioral findings using a computational model inspired by Rescorla-Wagner model. In essence, the model proposes that forgetting is a form of adaptive learning that can be updated based on prediction error rules in which engram relevancy is altered in response to environmental feedback.

We thank the Reviewer for summarizing the scope and depth of our manuscript, and for recognizing our efforts. We engage below the Reviewer’s specific criticisms of our interpretations.

Strengths:(1) The data presented in the current paper are consistent with the authors claim that seemingly forgotten engrams sometimes remain accessible. This suggests that retrieval deficits can lead to memory impairments rather than a loss of the original engram (at least in some cases).

We thank the Reviewer for their positive summary.

(2) The experimental procedures and statistics are appropriate, and the behavioral effects appear to be very robust. Several key effects are replicated multiple times in the manuscript.

We thank the Reviewer for their positive comments.

Weaknesses:(1) My major issue with the paper is the forgetting model proposed in Figure 7. Prior work has shown that neutral stimuli become associated in a manner similar to conditioned and unconditioned stimuli. As a result, the Rescorla-Wagner model can be used to describe this learning (Todd & Homes, 2022). In the current experiments, the neutral context will become associated with the unpredicted objects during training (due to a positive prediction error). Consequently, the context will activate a memory for the objects during the test, which should facilitate performance. Conversely, any manipulation that degrades the association between the context and object should disrupt performance. An example of this can be found in Figure 5A. Exposing the mice to the context in the absence of the objects should violate their expectations and create a negative prediction error. According to the Rescorla-Wagner model, this error will create an inhibitory association between the context and the objects, which should make it harder for the former to activate a memory of the latter (Rescorla & Wagner, 1972). As a result, performance should be impaired, and this is what the authors find. However, if the cells encoding the context and objects were inhibited during the context-alone sessions (Figure 5D) then no prediction error should occur, and inhibitory associations would not be formed. As a result, performance should be intact, which is what the authors observe.What about forgetting of the objects that occurs over time? Bouton and others have demonstrated that retrieval failure is often due to contextual changes that occur with the passage of time (Bouton, 1993; Rosas & Bouton, 1997, Bouton, Nelson & Rosas, 1999). That is, both internal (e.g. state of the animal) and external (e.g. testing room, chambers, experimenter) contextual cues change over time. This shift makes it difficult for the context to activate memories with which it was once associated (in the current paper, objects). To overcome this deficit, one can simply re-expose animals to the original context, which facilitates memory retrieval (Bouton, 1993). In Figure 2D, the authors do something similar. They activate the engram cells encoding the original context and objects, which enhances retrieval.Therefore, the forgetting effects presented in the current paper can be explained by changes in the context and the associations it has formed with the objects (excitatory or inhibitory). The results are perfectly predicted by the Rescorla-Wagner model and the context-change findings of Bouton and others. As a result, the authors do not need to propose the existence of a new "forgetting" variable that is driven by negative prediction errors. This does not add anything novel to the paper as it is not necessary to explain the data (Figures 7 and 8).

We thank the reviewer for clearly explaining their concern about our model. We are very sorry that we did not sufficiently explain that our model is, in fact, based on the classic Rescorla-Wagner model. The key equation of the model that updates “engram strength” is equivalent to the canonical Rescorla-Wagner model that is commonly used in research on reinforcement learning and decision-making (105). One potential minor difference is that we crucially assume different learning rates for positive and negative prediction errors. However, this variant of the Rescorla-Wagner model is common in the computational literature and is generally not regarded as a qualitatively different kind of model. In fact, it allows us to capture that establishing an object-context association (after a positive prediction error) is faster than the forgetting process (through negative errors).

The other equations that are explained in detail in the Methods are necessary to simulate exploration behavior and render the model suitable for model fitting. Concerning exploration behavior, we use the softmax function, which is commonly used in combination with the Rescorla-Wager model, in order to translate the learned quantity (in our case, engram strength) into behavior (here exploration). The other equations are necessary to fit the model to the data (learning rate α and behavioral variability in exploration behavior).

Therefore, we fully agree with the reviewer that the Rescorla-Wagner can explain our empirical results, in particular by assuming that the different manipulations affect the strength of object-context associations, which, in turn, governs forgetting as behaviorally observed.

In our previous version of the manuscript, we only referred to the Rescorla-Wagner model directly in the Methods. But to make this important point clearer, we now refer to the origin of the model multiple times in the Results section as well. See lines 81, 386-393.

We also agree with the reviewer that the learning/forgetting process can be described in terms of changes in object-context associations (e.g., inhibitory associations after a negative prediction error). Therefore, we now explicitly refer to the relationship between updated object-context associations and forgetting and highlight that we believe that stronger associations signal higher engram “relevancy”. See lines 386-393.

We have extended Figure 7 (new panels a and b), where we illustrate the idea that (a) object-context associations govern forgetting and (b) show the key Rescorla-Wagner equation, including a simple explanation of the main terms (engram strength, prediction error, and learning rate). Finally, we have also extended our discussion of the model, where we now directly state that the Rescorla-Wagner model captures the key results of our experiments. See lines 573-580.

In order to further support a link between our empirical data and computational modeling, we also added extra experiments that showed the modulation of engram cells within the dentate gyrus can regulate these object-context associations. See Supplementary Figure 12a-f and lines 401-404.

To summarize our reply, we agree with the reviewer’s comment and hope that we have clarified the direct relationship to the Rescorla-Wagner model.

(2) I also have an issue with the conclusions drawn from the enriched environment experiment (Figure 3). The authors hypothesize that this manipulation alleviates forgetting because "Experiencing extra toys and objects during environmental enrichment that are reminiscent of the previously learned familiar object might help maintain or nudge mice to infer a higher engram relevancy that is more robust against forgetting.". This statement is completely speculative. A much simpler explanation (based on the existing literature) is that enrichment enhances synaptic plasticity, spine growth, etc., which in turn reduces forgetting. If the authors want to make their claim, then they need to test it experimentally. For example, the enriched environment could be filled with objects that are similar or dissimilar to those used in the memory experiments. If their hypothesis is correct, only the similar condition should prevent forgetting.

We thank the Reviewer for this alternative perspective on our findings. First of all, we agree that this statement is speculative. The effects of enrichment on neural plasticity are well established and it likely contributes to the enhanced memory recall. It is important to emphasize that this process of updating is not necessarily separate from enrichment-induced plasticity at an implementational level, but part of the learning experience within an environment containing multiple objects. The enrichment or, more generally, experience, may therefore enhance memory through the modification of activity of specific engram ensembles. The idea of enrichment facilitating memory updating is consistent with the results obtained by the reminder experiments and further supported by our analysis with the Rescorla-Wagner computational model, where experience updates the accessibility of existing memories, possibly through reactivation of the original engram ensemble.

We would like to further clarify that our explanation concerns the algorithmic level, in contrast to the neural level. Based on the computational analyses using the Rescorla-Wagner model and in line with the reviewer’s previous comment on the model, we believe that forgetting is governed by the strength of object-context associations (or engram relevancy). Our interpretation is that stronger associations signal that the memory or engram representation is important ("relevant") and should not be forgotten. Accordingly, due to a vast majority of experiences with extra cage objects in the enriched environment, mice might generally learn that such objects are common in their environment and potentially relevant in the future (i.e., the object-context association is strong, preventing forgetting). Our speculation of these results is to help unify our empirical data with the computational model.

We believe that the Reviewer's alternative explanation in terms of synaptic plasticity, spine growth is not mutually exclusive with the modelling work. It is possible that the computational mechanisms that we explore based on the Rescorla-Wagner model are neuronally related to the biological mechanisms that the reviewer suggests at the implementational level. Therefore, ultimately, the two perspectives might even complement each other. We have included additional discussion to clarify this point. See lines 510-546.

(3) It is well-known that updating can both weaken or strengthen memory. The authors suggest that memory is updated when animals are exposed to the context in the absence of the objects. If the engram is artificially inhibited (opto) during context-only re-exposures, memory cannot be updated. To further support this updating idea, it would be good to run experiments that investigate whether multiple short re-exposures to the training context (in the presence of the objects or during optogenetic activation of the engram) could prevent forgetting. It would also be good to know the levels of neuronal reactivation during multiple re-exposures to the context in the absence versus context in the presence of the objects.

We thank the Reviewer for their comments. We agree that additional experiments would be helpful to further support the idea of updating. We have performed additional experiments to test the idea that multiple short re-exposures to the training context, in the presence of objects prevents forgetting. In this paradigm, mice were repeatedly exposed to the original object pair (Supplementary Figure S5a). The results indicate that repeated reminder trials facilitate object memory recall (Supplementary Figure 5b&c). These data indicated that subsequent object reminders over time facilitates the transition of a forgotten memory to an accessible memory. See Supplementary Figure S5 and Lines 279-287.

(4) There are a number of studies that show boundary conditions for memory destabilization/reconsolidation. Is there any evidence that similar boundary conditions exist to make an inaccessible engram accessible?

The Reviewer asks an interesting question about boundary conditions and engram accessibility. Boundary conditions could indeed affect the degree of destabilization and reconsolidation, the salience or strength of the memory, as well as the timing of retrieval cues. Future models could focus on understanding the specific boundary conditions in which a memory becomes retrievable and the degree to which it is sufficiently destabilized and liable for updating and forgetting. We have included additional discussion on the potential role of boundary conditions for engram accessibility. See lines 661-666.

(5) More details about how the quantification of immunohistochemistry (c-fos, BrdU, DAPI) was performed should be provided (which software and parameters were used to consider a fos positive neurons, for example).

We have added additional information for the parameters of quantification of immunohistochemistry. See lines 796-809.

(6) Duration of the enrichment environment was not detailed.

We have highlighted the details for the environmental enrichment duration. See lines 756.

**Reviewer #3 (Public Review):**
Summary:The manuscript by Ryan and colleagues uses a well-established object recognition task to examine memory retrieval and forgetting. They show that memory retrieval requires activation of the acquisition engram in the dentate gyrus and failure to do so leads to forgetting. Using a variety of clever behavioural methods, the authors show that memories can be maintained and retrieval slowed when animals are reared in environmental enrichment and that normally retrieved memories can be forgotten if exposed to the environment in which the expected objects are no longer presented. Using a series of neural methods, the authors also show that activation or inhibition of the acquisition engram is key to memory expression and that forgetting is due to Rac1.

We thank the Reviewer for summarizing the scope and depth of our manuscript, and indeed for recognizing our efforts. We engage below the Reviewer’s specific criticisms of our interpretations.

Strengths:This is an exemplary examination of different conditions that affect successful retrieval vs forgetting of object memory. Furthermore, the computational modelling that captures in a formal way how certain parameters may influence memory provides an important and testable approach to understanding forgetting.The use of the Rescorla-Wagner model in the context of object recognition and the idea of relevance being captured in negative prediction error are novel (but see below).The use of gain and loss of function approaches are a considerable strength and the dissociable effects on behaviour eliminate the possibility of extraneous variables such as light artifacts as potential explanations for the effects.

We thank the Reviewer for their positive comments.

Weaknesses:Knowing what process (object retrieval vs familiarity) governed the behavioural effect in the present investigation would have been of even greater significance.

The Reviewer touches on an important issue of the object recognition task. Understanding how experience alters object familiarity versus object retrieval and its impact on learning would help to develop better models of object memory and forgetting. We have added additional discussion. See lines 666-669.

The impact of the paper is somewhat limited by the use of only one sex.

We agree that using only male mice limits the impact of the paper. Indeed, the field of behavioural neuroscience is moving to include sex as a variable. Future experiments should include both male and female mice.

While relevance is an interesting concept that has been operationalized in the paper, it is unclear how distinct it is from extinction. Specifically, in the case where the animals are exposed to the context in the absence of the object, the paper currently expresses this as a process of relevance - the objects are no longer relevant in that context. Another way to think about this is in terms of extinction - the association between the context and the objects is reduced results in a disrupted ability of the context to activate the object engram.

We thank the reviewer for their insightful comment on the connection between engram relevance and memory extinction. Lacagnina et al., demonstrated that extinction training suppressed the reactivation of a fear engram, while activating a second putative extinction ensemble (59). In another study, these extinction engram cells and reward cells were shown to be functionally interchangeable (92). Moreover, in a study conducted by Lay et al., the balance between extinction and acquisition was disrupted by inhibiting the extinction recruited neurons in the BLA and CN (93). These results suggested that decision making after extinction can be governed by a balance between acquisition and extinction specific ensembles (93). Together, this may suggest that in the present study, when mice are repeatedly exposed to the training context, the association between the context and the objects is reduced, resulting in a disrupted ability of the context to activate the object engram. Therefore, memory relevance and extinction may operate similarly to effect engram accessibility, and in essence ‘forgetting’ of object memories may be due to neurobiological mechanisms similar to that of extinction learning (4). We have included additional discussion on the link between our results and the extinction literature. See lines 642-654.

**Recommendations for the authors:**

**Reviewer #1 (Recommendations For The Authors):**
Additional measures that may help interpretation of and clarify data are:A minute-by-minute analysis for training and testing may provide insight about the learning rate and testing temporal dynamics that may shed light substantially on differential levels of exploration. This should be applied across figures and would support conclusions from models in Figures 7-8 as well.Locomotion/distance travelled measures.

We have included additional analysis for a minute-by-minute analysis of training and testing of the object memory test at 24 hr, 2 weeks as well as under the standard housing and enrichment conditions. The results further support the initial finding that novel object recognition is increased in mice that recall the object at 24 hr. Similarly, mice housed in the enriched housing initially explore the novel object more compared to the familiar object. See Supplementary Figure 1 and 2, as well as lines 103-105 and 211-213.

The appropriate control for the context exposure figure would be to expose to a novel context in one group and the acquisition/testing context for the other.

We agree with the reviewer that an additional control of a novel context would further support our findings. Indeed, this line of investigate may dove-tail with the other reviewer comments on the role of competing engrams and interference. Future work could investigate the degree to which novel contexts and multiple memories can affect the rate of forgetting through engram updating. We have included additional discussion. See lines 643 and 655. However, in our experience it is necessary to pre-expose mice to different contexts before object exposure (e.g. Autore et al ’23), in order to form discriminate object/context associations. Establishing such a paradigm for this study would be at odds with the established paradigms and schedules in this current study. Moreover, the possibility that the effect of object displacement on forgetting requires the familiar context, or not, does not impact the main conclusions of this study. However, we agree that it is a point for expansion in the future.

A control virus+light group vs simply a no-light condition.

For optogenetic experiments. Control mice underwent the same surgery procedure with virus and optic fibre implantation. However, no light was delivered to excite or inhibit the respective opsin. Previous papers have shown laser light delivered to tissue expressing an AAV-TRE-EYFP lacking an light-opsin does cause cellular excitation. We have clarified this in the text. See lines 726-729.

**Reviewer #2 (Recommendations For The Authors):**
Minor details:(1) In the pharmacological modification of Rac 1, please specify what percentage of DMSO was used to dissolve Rac1 inhibitor and correct the typo 'DSMO'

Rac1 inhibitor (Ehop016) was reconstituted and prepared in PBS with 1% Tween-80, 1% DMSO and 30% PEG. We have clarified this in the text and corrected the typo, thank you. See lines 767.

(2) In the penultimate paragraph there is a typo 'predication error'

This is now corrected. Thankyou.

**Reviewer #3 (Recommendations For The Authors):**
I was unable to find information on what the No Light group consisted of. Was there a control virus infused, were the animals implanted with optical fibres (in the presence or absence of a virus), were they surgical controls, etc?

For optogenetic experiments. No Light Control mice underwent the same surgery procedure with virus and optic fibre implantation. However, no light was delivered to excite or inhibit the respective opsin. We have clarified this in the text. See lines 726-729.

The discussion lacked specificity in places. For example, the idea of eluding to 'other variables' is somewhat vague (p. 21, middle paragraph). Some examples of what other variables could be relevant would be helpful in capturing what direction or relevance the model may have going forward.

We have expanded the discussion of other variables which might impact engram relevance and how the model might be developed moving forward. These may include, boundary conditions of destabilization and reconsolidation, the salience or strength of the memory as well as the timing of retrieval cues or updating experience. Future models could focus on understanding the specific boundary conditions in which a memory becomes retrievable and the degree to which it is sufficiently destabilized and liable for updating and forgetting. The role of perceptual learning on memory retrieval and forgetting may also be an avenue of future investigation. Understanding how experience alters object familiarity versus object retrieval and its impact on learning would also help to develop better models of object memory and forgetting. In the current study, only male mice were utilized. Therefore, future work could also include sex as a variable to fully elucidate the impact of experience on the processes of forgetting. See lines 642-669.

In the same paragraph (p. 21, middle paragraph) there is mention of multiple engrams and how they can compete. The authors reference Autore et al (2023), but I thought Lacagina did this really beautifully also in an experimental setting. This idea is also expressed in Lay et al. (2022). So additional references would further strengthen the authors argument here.

We thank the reviewer for the additional references for discussing engram competition. We have included these papers in the discission. See lines 642-654.

Relatedly, environmental enrichment was considered in terms of object relevance. I wonder if the authors may want to consider thinking about their results in terms of effects on perceptual learning.

Indeed, perceptual learning maybe playing a role in environmental enrichment. We have included additional discussion. See lines 666-669.